# Circulating tumor DNA reveals mechanisms of lorlatinib resistance in patients with relapsed/refractory ALK-driven neuroblastoma

Esther R. Berko[1,2,15], Gabriela M. Witek [1,3,4,15], Smita Matkar[1,15], Zaritza O. Petrova[5,6,15], Megan A. Wu[5,6], Courtney M. Smith[5,6], Alex Daniels[1], Joshua Kalna[1], Annie Kennedy[1], Ivan Gostuski[4], Colleen Casey [1], Kateryna Krytska[1], Mark Gerelus [1], Dean Pavlick[7], Susan Ghazarian[8,9], Julie R. Park[10], Araz Marachelian[8,9], John M. Maris [1,3], Kelly C. Goldsmith[11,12,13], Ravi Radhakrishnan [4,14], Mark A. Lemmon [5,6] & Yaël P. Mossé [1,3]

Activating point mutations in *Anaplastic Lymphoma Kinase (ALK)* have positioned *ALK* as the only mutated oncogene tractable for targeted therapy in neuroblastoma. Cells with these mutations respond to lorlatinib in pre-clinical studies, providing the rationale for a first-in-child Phase 1 trial (NCT03107988) in patients with ALK-driven neuroblastoma. To track evolutionary dynamics and heterogeneity of tumors, and to detect early emergence of lorlatinib resistance, we collected serial circulating tumor DNA samples from patients enrolled on this trial. Here we report the discovery of off-target resistance mutations in 11 patients (27%), predominantly in the RAS-MAPK pathway. We also identify newly acquired secondary compound *ALK* mutations in 6 (15%) patients, all acquired at disease progression. Functional cellular and biochemical assays and computational studies elucidate lorlatinib resistance mechanisms. Our results establish the clinical utility of serial circulating tumor DNA sampling to track response and progression and to discover acquired resistance mechanisms that can be leveraged to develop therapeutic strategies to overcome lorlatinib resistance.

Neuroblastoma is an embryonic solid tumor of the peripheral sympathetic nervous system that remains an often-lethal childhood cancer despite intensive cytotoxic therapies, and survivors are burdened with treatment-related comorbidities[1]. Neuroblastomas are characterized by extensive intra-tumoral and stroma-derived heterogeneity and harbor pre-existing and acquired subclonal populations that are postulated to confer therapy resistance. Heritable mutations in the *Anaplastic Lymphoma Kinase (ALK)* gene have been shown to be the major cause of familial neuroblastoma[2]. In addition, somatic mutations are currently known to occur in 14% of sporadic high-risk neuroblastomas[2–5]. Furthermore, relapsed neuroblastoma harbors an increased proportion of somatic mutations, with enrichment of *ALK* activating mutations compared to diagnostic tumors, with a frequency of 20% and rising as we sequence patient tumors and/or plasma more routinely at the time of relapse[6–9]. Oncogenic *ALK* mutations occur primarily at three hotspots, corresponding to positions F1174, F1245, and R1275 in the tyrosine kinase domain (TKD) of the full-length ALK receptor tyrosine kinase (RTK), and confer ligand-independent

tyrosine kinase activity[4,10]. These findings have positioned *ALK* as the only currently tractable oncogene for targeted therapy in neuroblastoma.

Inhibitors of *ALK* tyrosine kinase activity have been well studied in a different set of cancers that do not express intact ALK, but instead express oncogenic ALK fusion proteins in the cytoplasm, particularly non−small cell lung cancer (NSCLC)[11,12]. Patients with ALK fusion NSCLC are frequently treated with serial generations of ALK inhibitors, but resistance almost always develops−through acquired single point mutations in the *ALK* TKD in approximately one-third of cases[13]. Distinct patterns of mutations are seen for different ALK inhibitors[14], but all such single mutations appear to retain sensitivity to the potent third-generation ALK inhibitor, lorlatinib[15]. Patients with ALK fusion NSCLC who acquire compound (≥2) *ALK* TKD mutations during lorlatinib treatment, however, have been found to show markedly decreased sensitivity to lorlatinib and clinical progression[13,16−21].

In neuroblastoma, where intact ALK is activated by specific point mutations rather than aberrant fusions, the first-generation ALK inhibitor crizotinib was found to exhibit modest efficacy in ALK-driven xenograft models[10,22−26], and differential preclinical and clinical activity depending on the specific *ALK* driver mutations[10,22,27,28]. Not unexpectedly given these observations, only marginal activity was observed in early-phase clinical trials of crizotinib in patients with relapsed or refractory *ALK*-mutated neuroblastoma[27−29]−contrasting with the robust and sustained responses seen in patients with *ALK* fusion-driven tumors[30,31]. Importantly, lorlatinib was subsequently shown to overcome the de novo resistance of intact *ALK* variants seen in neuroblastoma to first and second-generation ALK inhibitors[23,24]. Lorlatinib was found to exert unprecedented preclinical activity as a single agent in neuroblastoma patient-derived xenografts with any of the three hotspot mutations[24]. This advance led to a first-in-child trial of lorlatinib for patients with ALK-driven refractory or relapsed high-risk neuroblastoma (NCT03107988) described elsewhere[32]. To track the evolutionary dynamics and heterogeneity of neuroblastoma, and to detect early emergence of resistance to lorlatinib, we collected serial circulating tumor DNA (ctDNA) via liquid biopsy from patients enrolled on this clinical trial−an approach that we and others recently showed can provide valuable real-time data on genomic evolution and development of resistance to therapy in neuroblastoma[33−35].

Here, we report the results of prospective serial ctDNA analysis from patients with relapsed or refractory ALK-driven neuroblastoma who received lorlatinib therapy. We identify genetic mechanisms of lorlatinib resistance, including off-target acquisition of mutations in the RAS-MAPK pathway and on-target acquisition of new in-cis compound mutations in *ALK*. We functionally validate the effect of compound *ALK* mutations in neuroblastoma using in vitro cell-based, biochemical, and computational approaches. Our results provide insight into acquired lorlatinib resistance mutations in patients with ALK-driven relapsed/refractory neuroblastoma. Importantly, our findings also provide the framework for developing new therapeutic strategies aimed at preventing emergence of resistance mutations and intervention strategies when they do emerge.

## Results
### Trial design and ctDNA baseline results
We present here correlative studies performed as part of the first-in-child New Approaches to Neuroblastoma Therapy (NANT) Consortium Phase 1 study (NANT2015-02) of lorlatinib in children, adolescents, and adults with ALK-driven refractory/relapsed neuroblastoma[32]. Primary aims of the trial were to determine the toxicity, pharmacokinetics, and recommended phase 2 dose (RP2D) of lorlatinib administered both as monotherapy and in combination with topotecan/cyclophosphamide. A secondary aim was to evaluate the anti-tumor activity by determining response rate. An additional exploratory aim, on which we report here, was to prospectively determine the frequency of ctDNA

detection of *ALK* and other acquired mutations both at study entry and when each disease evaluation was performed. To identify and trace the progression of mechanisms of resistance to lorlatinib, we used the FoundationOne Liquid Assay[36] to sequentially profile ctDNA from all patients with *ALK*-mutated relapsed/refractory neuroblastoma enrolled on the NANT phase 1 clinical trial NCT03107988. This trial enrolled 49 patients between September 2017 and February 2022. Optional blood sampling was prospectively performed at predefined time points including pretreatment, after courses 2,4,6, and then after every 4th course and at disease progression. Patients <18 years of age were treated across 5 dose levels (cohort A1). Patients ≥18 years of age were treated across 2 dose levels (cohort A2), and patients who received lorlatinib in combination with chemotherapy (cohort B2) were treated across two dose levels as described[32]. Twenty patients (41%) had previously been treated with earlier-generation ALK inhibitors, and 29 (59%) were ALK inhibitor naïve.

A total of 46 patients were evaluable for the ctDNA study. Thirty-nine had a sample profiled at enrollment, and 32 (82%) had detectable *ALK* mutations in the pre-therapy sample: 10 at R1275, 15 at F1174, and 4 at F1245. One patient harbored *ALK* amplification plus an F1174 mutation, and one had both an *ALK* ΔD1276-R1279InsE indel and a G1202R mutation. The remaining patient (patient 46) harbored all 3 hotspot mutations and came off study due to progressive disease prior to start of therapy. The overall *ALK* variant allele frequency (VAF) in ctDNA samples at enrollment ranged from 0% to 63% (median 4.37), as denoted by the circle size in Fig. 1 (see also Source Data). The detectable *ALK* VAF at enrollment did not directly correlate with disease burden as measured by total Curie score (Supplementary Data Fig. 1). A total of 41 patients had more than one serial sample profiled and were therefore eligible for further analysis (Supplementary Data Fig. 2 and Source Data). The treatment status of each, as well as whether they were ALK inhibitor naïve, were receiving lorlatinib monotherapy, or were receiving combination chemotherapy (in Cohort B2) is denoted in Fig. 1.

### Circulating tumor *ALK* VAF varies with clinical response to lorlatinib
Patients segregated into distinct groups based on their ctDNA *ALK* VAF levels and how/whether they responded to lorlatinib therapy. Over half (21/41; 51%) clustered into an 'ALK Dependent' Group 1, in which mutated *ALK* VAF levels correlated with clinical response (Fig. 1a). In most Group 1 patients (patients 1, 2, 5, 6, 13, 14, 17, 24, 25, 30, 31, 36, 40, 41, 43, 45, and 49), *ALK* VAF decreased or remained stable with initial clinical response to lorlatinib and/or increased with disease progression. Patient 1 additionally harbored *ALK* amplification that remained detectable throughout therapy. In a second subset within Group 1, disease progression occurred while on therapy (patients 16, 26, 28, 32), and *ALK* VAF remained persistently high. In these patients, the estimated proportion of detectable ctDNA in the cell-free DNA comprehensive tumor fraction (CTF) followed the same trend as the *ALK* VAF (Supplementary Data Fig. 3).

A second 'ALK Independent' Group 2 (Fig. 1b) contained eleven of the 41 patients (27%), characterized by a lack of correlation between circulating tumor *ALK* VAF and clinical response. Most of these patients showed a decrease in *ALK* VAF despite disease progression, suggesting underlying tumor heterogeneity and outgrowth of subclones without an *ALK* aberration. In four such patients (7, 15, 27, and 37), alternative (non-*ALK*) mutations were also detectable at enrollment and were enriched for at disease progression (Supplementary Data Fig. 4). In two of these patients with detectable high CTF (27 and 37), the presence of the alternative mutation was identified. Patient 27 showed an *NF1 E163** (VAF = 32.8%) mutation in the enrollment ctDNA sample (alongside *MYCN* amplification) plus a low level of *ALK* F1174L (VAF = 0.32%). The sample taken from this patient at disease progression retained the *MYCN* amplification and showed enrichment of only

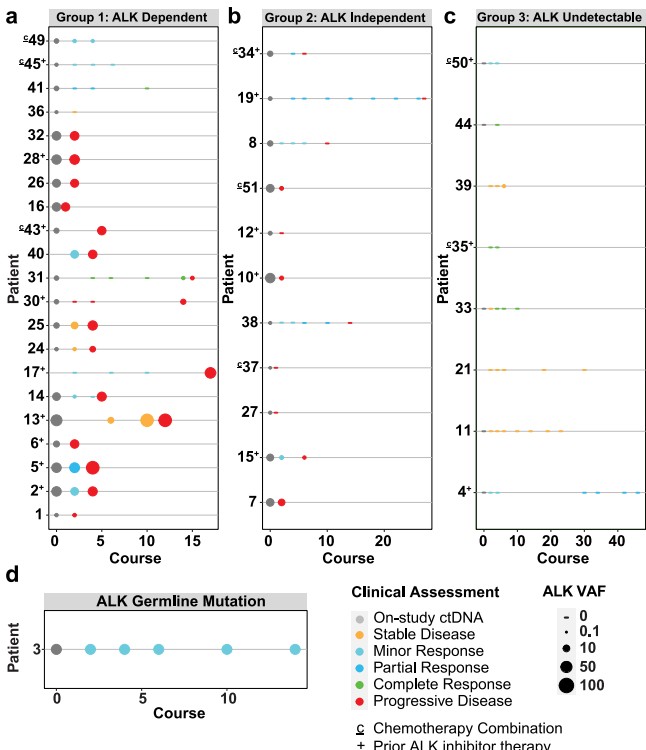

**Fig. 1 | Circulating plasma *ALK* VAF varies with clinical response to therapy. a** In Group 1, circulating *ALK* VAF correlated with disease response, as patients' tumors remained dependent on mutated activated ALK. In patients 1, 2, 5, 6, 13, 14, 17, 24, 25, 30, 31, 40, and 43, *ALK* VAF decreased with clinical response to lorlatinib and subsequently increased with disease progression. In patients 16, 26, 28, and 32, *ALK* VAF remained elevated as patients had progressive disease despite treatment. In patients 36, 41, 45, and 49 *ALK* VAF decreased with ongoing response to lorlatinib therapy. **b** In Group 2, circulating *ALK* VAF did not correlate with tumor response, decreasing despite disease progression–suggesting potential emergence of alternative oncogenic drivers of tumor growth. **c** In Group 3, circulating *ALK* was never detectable, and patients demonstrated persistent response to therapy (with the exception of sample 3 in patient 39 where minimal *ALK* VAF was detected at 0.18%). **d** Patient 3 harbored a germline *ALK* R1275Q mutation that remained stable at the expected 50% VAF, with complete response to lorlatinib. Patient numbers are annotated with a 'c' if they were receiving combination lorlatinib/chemotherapy, and with a '+' sign if they had received prior ALK inhibitor therapy. All VAF values are given as 'SV percent reads' in Source Data. VAF variant allele frequency, ctDNA circulating tumor DNA. Source data are provided as a Source Data file.

the *NF1* variant (VAF = 66.9%)–with no detectable *ALK* mutation. Interestingly, patient 27 had a mixed clinical response to lorlatinib, with some soft tissue tumors completely resolving while other lesions progressed on therapy. When multiple progressing tumors in this patient were biopsied and sequenced, they no longer showed mutated *ALK*, but retained the *NF1* mutation. These observations suggest that lorlatinib resistance in this patient was due to *ALK*-independent clones, validating the ability of ctDNA analysis to detect this resistance mechanism. Other mutations in this subset of four patients were identified in *TP53*, *HRAS*, and *MTOR* (Supplementary Data Fig. 4). In four additional Group 2 patients (patients 10, 12, 38, and 51), *ALK* mutations detected at enrollment were found to be decreased at disease progression, but no alternative genetic driver could be identified –suggesting epigenetic or other molecular mechanisms of resistance. For the remaining three patients in ctDNA Group 2 (patients 8, 19, 34), *ALK* VAF decreased from enrollment along with initial clinical response, followed eventually by progressive disease. *ALK* mutations were no longer detectable in these patients, but we could also not identify the genetic drivers of resistance as no ctDNA was detectable in

these patients at disease progression despite increased disease burden.

Eight patients with serial ctDNA samples fit into a third 'ALK Undetectable' Group 3 (Fig. 1c), as *ALK* mutations were not detected in ctDNA samples. For patient 39, the *ALK* F1174L mutation (VAF = 0.18%) was detectable after course 6 of therapy, coinciding with continued stable disease on radiographic evaluation. This patient went on to have an objective response to lorlatinib, but no subsequent ctDNA samples were sent. All patients in Group 3 responded to therapy, with duration of response ranging from 4 to 48 cycles of therapy; all hotspot mutations were represented, and none harbored *MYCN* amplification. No impact of cohort, dose level, or prior ALK inhibitor treatment was seen on ctDNA response to lorlatinib treatment. In some of these patients (4, 21, 33, 35, and 50), CTF was intermittently detectable at low levels. Finally, patient 3 harbored a germline *ALK* R1275Q mutation that remained stable at the expected 50% VAF (Fig. 1d); this patient discontinued therapy after 18 months, has been in remission for 5 years, and continues to be monitored by ctDNA analysis.

### Emerging off-target mutations as drivers of acquired resistance to lorlatinib

We next screened for genetic drivers of acquired resistance in our patient cohort and observed potential off-target mechanisms in eleven patients (11/41, 27%). These are depicted in Fig. 2 (annotated in red) as alterations that arise coincident with disease progression. For example, patient 17 developed an *FGFR1* N546K variant (VAF = 60.4%) with disease progression, and patient 43 acquired both an *MLL2* W2818* mutation (VAF = 30%) and a *CIC* S554* mutation (VAF = 12.8%). Twelve patients harbored downstream RAS-MAPK/PI3K pathway aberrations (Fig. 3). In three of these (patients 7, 27, and 37), none of which had received prior ALK inhibitor treatment, the RAS-MAPK/PI3K pathway alterations were seen at enrollment and no response to lorlatinib therapy was observed, with progression at timepoint 2 (Supplementary Data Fig. 4). In patient 7, low-level gains of unclear functional significance in *MET*, *BRAF*, *CDK6* and *EGFR* were observed in the enrollment sample. In the second sample, sent at timepoint 2 with disease progression, high level amplification of *MET* and *BRAF* was detected, suggesting potential selection of these alterations in the resistant cancer clone. Six patients (2, 13, 14, 26, 31, 32) acquired new *RAS-MAPK/PI3K* pathway alterations during treatment, nearly always preceding clinical/radiographic disease progression. Alterations of this type were seen in patients with any of the three baseline *ALK* hotspot mutations, and regardless of drug dose level or *MYCN* status. Two of these had prior treatment with ALK inhibition (patient 2 with crizotinib and ceritinib, patient 13 with crizotinib and alectinib), and the remainder did not. Interestingly, in three patients (patients 2, 13, 14) the acquisition of *RAS-MAPK* pathway alterations (in *PTPN11* or *BRAF*) coincided with gain of a secondary *ALK* mutation (Fig. 3) and may contribute to lorlatinib-resistant disease. Patient 26 acquired multiple NF1 indels (all with VAF < 1%), patient 31 developed an *HRAS* G13R mutation (VAF = 0.14%), and patient 32 acquired a *PIK3CA* H1047R mutation (VAF = 0.18%), all preceding disease progression. These findings argue that serial analysis of ctDNA samples as described here could have value in identifying other actionable mutations at or before disease progression during ALK-targeted therapy.

### Acquisition of secondary *ALK* TKD mutations with disease progression

Six patients were found to have acquired secondary mutations in the *ALK* TKD at the time of disease progression (Fig. 4, Supplementary Note 1), contending that compound *ALK* mutations can cause lorlatinib resistance despite the potency of this inhibitor. For patients with a baseline F1174 *ALK* mutation, we were able to determine if these secondary mutations occurred in *cis* (on the same allele) or in *trans* with the activating *ALK* hotspot mutation, as the mutations are in close

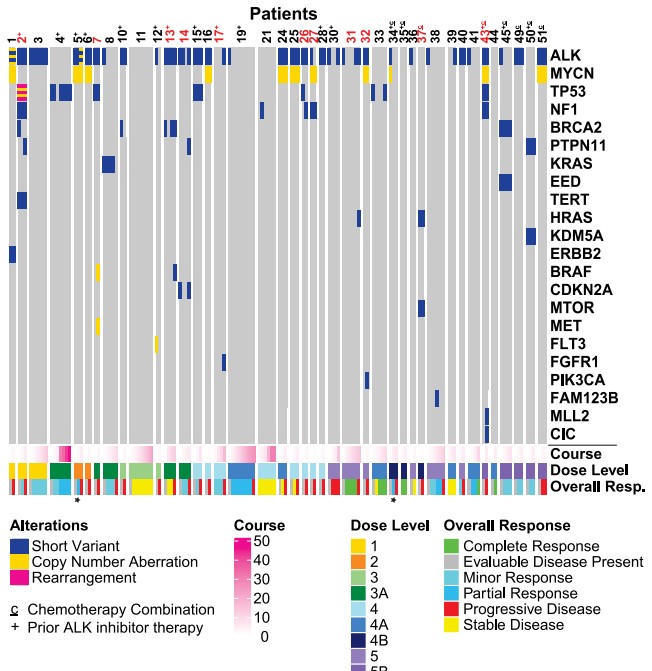

**Fig. 2 | Landscape of detectable circulating tumor DNA genetic mutations during lorlatinib treatment.** An Oncoplot representation demonstrates mutations detectable at enrollment and throughout treatment in each patient. Each column shows the pathogenic or likely pathogenic alterations found in a single ctDNA sample from a study patient, with patient number denoted above (those annotated in red have acquired off-target genetic mechanisms of lorlatinib resistance). The corresponding treatment course, dose level, and clinical response at the time of genetic profiling for each sample is color coded below. Asterisks indicate samples in which radiography could not be incorporated as part of clinical evaluation. Patient numbers are annotated with a 'c' if they were receiving combination lorlatinib/chemotherapy, and with a '+' sign if they had received prior ALK inhibitor therapy. Source data are provided as a Source Data file.

proximity and are therefore both included in individual short sequencing reads. In all but one assessable case (patient 24), the secondary mutation did occur in *cis* with the F1174 mutation, sometimes with multiple secondary mutations each occurring in *cis* with the baseline mutation but in *trans* with each other (e.g., patient 5 in Fig. 4). Thus, compound F1174L/G1202R, F1174L/D1203N, F1174C/G1202R and F1174L/L1196M mutations were all observed in the ctDNA of patients at the time of disease progression on lorlatinib (Fig. 4). It is important to note that F1174 mutations—which potently activate ALK in neuroblastoma[10]—also appear in several compound mutations associated with lorlatinib resistance in NSCLC[17,19,21]. For neuroblastoma patients with *ALK* F1245 or R1275 hotspot mutations (Fig. 4), genomic distance precludes assignment of secondary mutations as *cis* versus *trans* based on our sequencing reads. Mutations at these hotspot positions have not been seen in patients with *ALK* fusions who developed lorlatinib-resistant compound mutations.

In all 6 of the patients summarized in Fig. 4 (three of whom also harbored *MYCN* amplification) the secondary *ALK* mutation was acquired after a period of initial disease response to lorlatinib, and its appearance preceded (or coincided with) disease progression. Three of these patients had received prior treatment with ALK inhibitors; patient 2 had received crizotinib and ceritinib, patient 5 had received crizotinib, and patient 13 had received crizotinib and alectinib. Prior treatment with ALK inhibitors did not appear to affect either time to development of compound mutations or response to lorlatinib. Importantly, none of the 11 patients (27% of the cohort) who received therapy at RP2D of lorlatinib for children <18 years of age (115 mg/m²— marked purple in the key to Fig. 2) acquired compound *ALK* mutations.

In addition to these six patients with *ALK* hotspot mutations, patient 19 instead had a ΔD1276-R1279InsE *ALK* deletion (VAF = 1.1%), replacing residues D1276 to R1279 in the ALK TKD activation loop with glutamate, which we show below is activating. In addition, a G1202R mutation (VAF = 0.15%) was detected at enrollment (which disappeared at the end of course 2 of therapy). This adult patient had previously received 3 years of crizotinib with stable disease, and then progressed prior to enrollment on the lorlatinib trial when the additional G1202R mutation was detected in ctDNA. This patient had a complete metabolic and partial anatomic response to lorlatinib, remained on protocol therapy for 27 courses, and is now on commercial supply of lorlatinib. No further pathogenic (or *ALK*) mutations were detected in this patient's ctDNA samples.

## Engineered compound mutations in *ALK* cause resistance to lorlatinib in neuroblastoma cell lines

We next asked whether the newly acquired compound mutations described above directly cause resistance to lorlatinib. As mentioned, G1202R and D1203N mutations were previously seen in combination with F1174 mutations in lorlatinib-resistant ALK fusion-positive lung cancer[17,19,21,37]. The (*cis*) F1174L/L1196M compound mutation seen in patient 25 was also previously detected when lorlatinib resistance of an ALK fusion was modeled in vitro[13]. We modeled in *cis ALK* compound mutations in human neuroblastoma-derived cell lines to assess their effects on lorlatinib pharmacodynamics. We used CRISPR-CAS9 with an ouabain co-selection system to enrich for homology-directed repair (HDR) editing[38] (Supplementary Data Fig. 5a) to introduce L1196M and G1202R mutations individually into the endogenous F1174L-mutated *ALK* gene harbored by the Kelly cell line (Supplementary Data Fig. 5b) and into the R1275Q-mutated *ALK* gene in the CHLA-20 cell line (Supplementary Data Fig. 5c). CRISPR knock-in editing in *cis* was confirmed in single-cell colonies by TA cloning followed by Sanger sequencing of individual colonies (Supplementary Data Fig. 5a). For the Kelly-derived cells, those expressing $ALK^{F1174L}$ (parental), $ALK^{F1174L/L1196M}$ and $ALK^{F1174L/G1202R}$ mutations showed significantly different responses to lorlatinib when cell viability was determined after 120 h of treatment with different doses of lorlatinib (Fig. 5a). Compared with an $IC_{50}$ of $35 \pm 6$ nM for Kelly cells, $IC_{50}$ was increased by ~50-fold for the F1174L/L1196M compound mutation ($IC_{50} = 1736 \pm 877$ nM: $P < 0.0001$) and >10-fold for the F1174L/G1202R compound mutation ($IC_{50} = 394 \pm 52$ nM: $P < 0.0001$). Similarly, the lorlatinib $IC_{50}$ of $15 \pm 4$ nM in CHLA cells was increased to $277 \pm 57$ nM ($P < 0.0001$) when the *cis* L1196M mutation was introduced, and to $318 \pm 58$ nM ($P < 0.0001$) when the *cis* G1202R mutation was introduced (Fig. 5b), a ~20-fold increase. These data argue that compound mutations including the F1174 *ALK* hotspot mutations are relevant for lorlatinib resistance in ALK-driven neuroblastoma.

## *ALK* compound mutations directly reduce lorlatinib sensitivity in vitro

To understand how these compound mutations cause lorlatinib resistance, we also conducted in vitro biochemical studies of the purified ALK tyrosine kinase domain (TKD). Variants harboring *cis* F1174L/G1202R, F1174L/L1196M, R1275Q/G1202R or R1275Q/L1196M compound mutations were expressed and purified. As shown in Fig. 5c, d and Supplementary Table 1, adding a G1202R mutation in *cis* to F1174L- or R1275Q-mutated ALK-TKD increased the apparent $IC_{50}$ for lorlatinib inhibition by more than 10-fold ($P = 0.028$ for F1174L; $P = 0.038$ for R1275Q), in agreement with our cellular results. Notably, a secondary G1202R mutation renders F1174L/G1202R and R1275Q/G1202R compound forms of the ALK TKD no more sensitive to lorlatinib than singly mutated F1174L or R1275Q ALK variants are to crizotinib, consistent with a loss of clinical response (Fig. 5c, d and Supplementary Table 1). Introducing the L1196M mutation into F1174L-mutated ALK-TKD increased apparent $IC_{50}$ to a smaller extent (~5-fold; $P = 0.025$) than

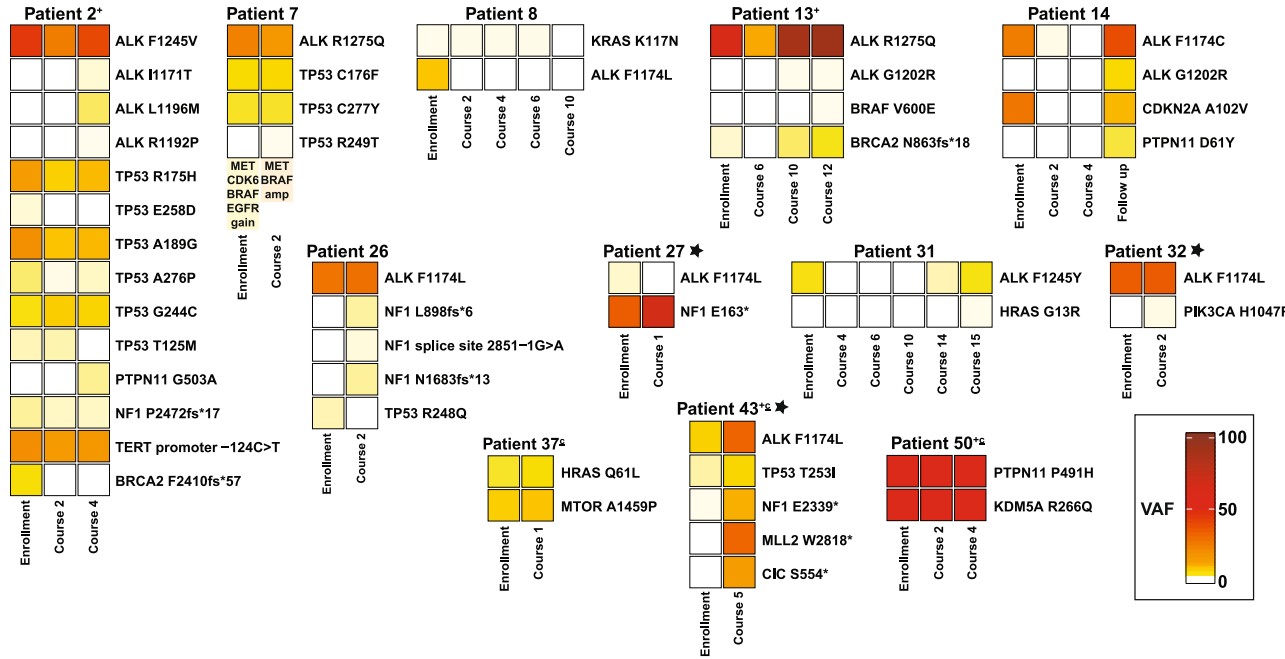

**Fig. 3 | Emergence of new *RAS-MAPK* mutations during lorlatinib treatment.** VAFs for *ALK* mutations and mutations in *RAS-MAPK* pathway genes during lorlatinib treatment are listed, all corresponding with disease progression. Boxes are colored by VAF according to the scale shown. Patient numbers are annotated with a '+' sign if they had received prior ALK inhibitor therapy, a 'c' if they were receiving combination lorlatinib/chemotherapy. Asterisks indicate patients with *MYCN* amplification. VAF variant allele frequency. Source data are provided as a Source Data file.

introducing G1202R—a difference that is reversed in the Kelly cell experiments, possibly because of effects of the secondary mutations on ALK protein expression or ALK activity. For ALK-TKD activated by the R1275Q mutation (Fig. 5d), the secondary L1196M mutation reduces lorlatinib sensitivity by just ~2.8-fold ($P = 0.17$) when measured as a function of in vitro kinase activity. The greater effect of the L1196M mutation on lorlatinib resistance in cellular settings than in vitro may result from the fact that this mutation also increases catalytic activity[10] and therefore (presumably) transforming ability. Note the increase in $k_{cat,app}/K_{M,ATP}$ for the F1174L/L1196M and R1275Q/L1196M variants as listed in Supplementary Table 1.

The increased IC50 caused by these secondary *ALK* mutations does not appear to reflect altered ATP binding as described for differential crizotinib sensitivity of singly-mutated ALK variants in neuroblastoma[22,24]. $K_{M(ATP)}$ is significantly smaller for F1174L-mutated ALK-TKD than for R1275Q[22], reducing crizotinib sensitivity. This difference was maintained in our assays ($P = 0.01$) but adding the G1202R or L1196M mutation in *cis* did not significantly alter $K_{M(ATP)}$ in either case (Fig. 5e). Instead, these secondary mutations appear to increase the estimated inhibition constant ($K_i$)[39] for lorlatinib by >10-fold (Supplementary Table 1), consistent with the observed resistance. As shown in Supplementary Data Fig. 6a, b and Supplementary Table 1, the variants with compound mutations are also substantially less sensitive to crizotinib than the parental F1174L and R1275Q variants.

We also studied the unusual ΔD1276-R1279InsE *ALK* mutation seen in patient 19 to confirm whether this alteration causes constitutive ALK activation. As shown in Fig. 5f, the ΔD1276-R1279InsE variant robustly promotes focus formation in NIH 3T3 cells, with an apparently stronger transforming ability than F1174L or R1275Q. In addition, biochemical measurements showed that the unphosphorylated ΔD1276-R1279InsE-mutated ALK-TKD is constitutively active, with a $k_{cat}$ value of $71 \pm 17$ min$^{-1}$ (Supplementary Data Fig. 6c, d), compared with just 9 min$^{-1}$ for wild-type, 365 min$^{-1}$ for F1174L and 119 min$^{-1}$ for R1275Q[10]. The fact that patient 19 progressed on crizotinib suggests that—in the context of ΔD1276-R1279InsE as a driver—the G1202R mutation seen on enrollment in the current trial causes resistance to crizotinib but not to lorlatinib.

## Structural mechanisms for lorlatinib resistance of neuroblastoma compound mutations

We next used molecular dynamics (MD) simulations to model the effects of the compound mutations (F1174L/L1196M, F1174L/G1202R, R1275Q/L1196M, and R1275Q/G1202R) on ALK-TKD, and used induced fit docking (IFD) to assess lorlatinib interaction energies. Docking of lorlatinib to wild-type or singly mutated ALK-TKD agreed well with the binding configuration seen in an ALK-TKD/lorlatinib crystal structure (Fig. 6a, Supplementary Data Fig. 7a–c). Relative docking scores (from IFD) calculated for hot spot-mutated ALK variants also correlated reasonably well ($R^2 = 0.86$) with experimentally-derived $-\log(K_i)$ values estimated from our earlier IC50 measurements for lorlatinib inhibition[24] (Supplementary Data Fig. 7d). Extending these IFD calculations to the compound mutations indicated that the second mutation reduces lorlatinib docking score for each compound mutation except the F1174L/G1202R combination (Fig. 6b), which showed wide variation associated with conformational heterogeneity (Supplementary Data Fig. 7e, f and Supplementary Table 3).

Our in silico studies suggest that the L1196M (gatekeeper) mutation decreases docking score through small structural changes that cause steric hindrance (Fig. 6c, and Supplementary Data Fig. 7g), as described previously for crizotinib resistance in *ALK* fusion lung cancer[40]. These small structural changes are sufficient for the L1196M mutation alone to cause crizotinib resistance[13], but are overcome by the high potency of lorlatinib[13,15]. When an F1174 mutation is also present, however, the increased ATP-binding affinity of ALK-TKD[10,22] (Fig. 5e) appears to effectively reduce lorlatinib potency and allow the L1196M mutation to cause resistance. Indeed, an F1174L/L1196M compound mutation was previously reported to promote lorlatinib resistance in a Ba/F3 cell model with an *EML4-ALK* fusion[13].

Interestingly, the G1202R mutation appears to promote lorlatinib resistance through a distinct mechanism, which differs for F1174L/G1202R and R1275Q/G1202R compound mutations. In the R1275Q/

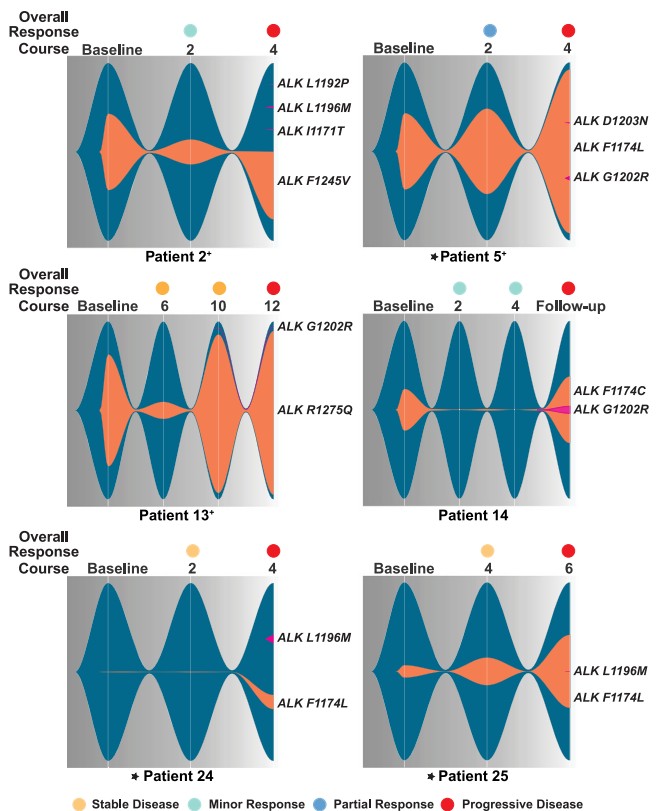

**Fig. 4 | Development of compound *ALK* mutations corresponds with tumor resistance and disease progression.** Fishplots demonstrate detectable *ALK* at each timepoint, with dark blue background and orange clones showing the original neuroblastoma *ALK* mutation and the size of the fish proportional to VAF (see Source Data). Secondary *ALK* mutations are colored magenta; when occurring in *cis* they are depicted within the orange clone and when in *trans* or unknown, they are depicted outside the clone. Asterisks indicate patients with *MYCN* amplification, and patient numbers are annotated with a '+' sign if they had received prior ALK inhibitor therapy. Source data are provided as a Source Data file.

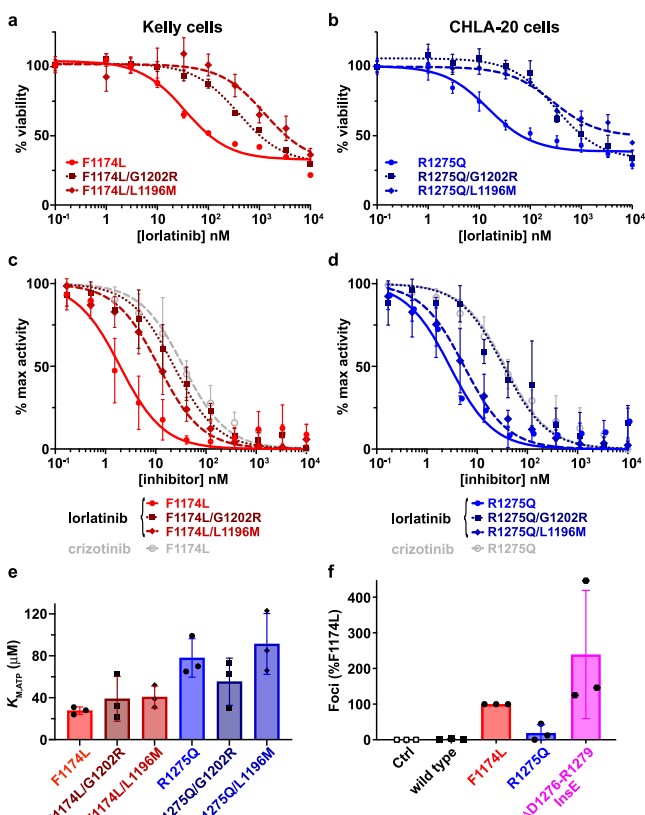

**Fig. 5 | Cellular and biochemical studies of *ALK* compound mutations. a** Cell viability assays at different lorlatinib concentrations of Kelly cells harboring either the single parental *ALK* F1174L driver mutation (red circles, solid curve) or the compound mutations F1174L/L1196M (medium red diamonds, dashed curve), or F1174L/G1202R (dark red squares, dotted curve) introduced in *cis* using CRISPR. **b** Cell viability assays at different lorlatinib concentrations of CHLA-20 neuroblastoma cells harboring either the single parental *ALK* R1275Q mutation (blue circles, solid curve) or the compound mutation F1174L/G1202R (dark blue squares, dotted curve). Data are plotted as the mean ± SD of three biological replicates, each performed in technical triplicate. **c** Comparison of in vitro inhibition of purified ALK-TKD for different F1174L-based variants. IC$_{50}$ values for F1174L-mutated ALK-TKD were assessed for lorlatinib (red circles, solid red curve: IC$_{50}$ = 2.3 ± 1.1 nM) and crizotinib (open gray circles, dashed gray curve: IC$_{50}$ = 40 ± 20 nM), and compared with lorlatinib IC$_{50}$ values for F1174L/L1196M (medium red diamonds, dashed curve: IC$_{50}$ = 12 ± 6.2 nM) and F1174L/G1202R (dark red squares, dotted curve: IC$_{50}$ = 26 ± 16 nM). **d** Comparison of in vitro inhibition of ALK-TKD for different R1275Q-based variants. IC$_{50}$ values for R1275Q-mutated ALK-TKD were assessed for lorlatinib (blue circles, solid blue curve: IC$_{50}$ = 2.9 ± 0.8 nM) and crizotinib (open gray circles, dashed gray curve: IC$_{50}$ = 38 ± 24 nM), and compared with lorlatinib IC$_{50}$ values for R1275Q/L1196M (medium blue diamonds, dashed curve: IC$_{50}$ = 8 ± 5.2 nM) and R1275Q/G1202R (dark blue squares, dotted curve: IC$_{50}$ = 40 ± 21 nM). **e** Measured $K_{M,ATP}$ values for different ALK-TKD variants, with numbers tabulated in Supplementary Table 1. **f** Focus formation assay results for ΔD1276-R1279InsE ALK (magenta) compared with wild-type (black), F1174L (red), and R1275Q (blue). Data are plotted as the mean ± SD of three biological replicates, each performed in technical duplicate. Source data are provided as a Source Data file.

G1202R variant, the side chain of the arginine that replaces G1202 projects towards solvent but increases the bulk close to the bound lorlatinib (Fig. 6d; Supplementary Data Fig. 7h) to cause a slight upward displacement of the inhibitor (and other small structural changes) and reduce lorlatinib docking score (Fig. 6b). Similar conclusions were drawn by Yoda et al.[13] from MD studies of the lorlatinib-resistant L1196M/G1202R compound mutation seen in *ALK* fusion-driven lung cancer. Unexpectedly, our modeling of the F1174L/G1202R compound mutation suggested a wide range of docking scores (Fig. 6b). Our MD analysis argues that, rather than simply impairing docking of lorlatinib through structural changes, the F1174L/G1202R variant spends a significant amount of time in a conformation with the arginine side chain at position 1202 projected into and fully occluding the lorlatinib binding site (Fig. 6e). We refer to this as the 'gate-closed' conformation, in which binding of lorlatinib, but not ATP, is prevented (Fig. 6e). This gate-closed conformation occurs frequently (>80%) in simulations of ALK-TKD with either a single G1202R mutation or the F1174L/G1202R compound mutation (Fig. 6f), but much more rarely in other variants studied here—which primarily assume the 'gate-open' conformation.

**Kinetic model for lorlatinib resistance in mutated ALK**
We also modeled the ability of lorlatinib to compete with ATP for binding to ALK-TKD in the different compound mutations, with- and without including the alternating gate-closed/gate-open conformations (Supplementary Data Fig. 8). We derived relative rates of binding

and dissociation of ATP from experimental $K_{M,ATP}$ values, and of lorlatinib from calculated lorlatinib docking scores that we calibrated to experimental estimates of $K_i$ and thus $K_D$ for R1275Q and F1174L variants; see Supplementary Note 2). Increases in IC$_{50}$ caused by the L1196M mutation could readily be accounted for by a simple competition model with ALK-TKD in the gate-open conformation (Fig. 6g, h and Supplementary Data Figs. 8a, c, e). Resistance of the F1174L/G1202R variant, by contrast, could only be accounted for when the gate-closed conformation was expressly included (Fig. 6h and

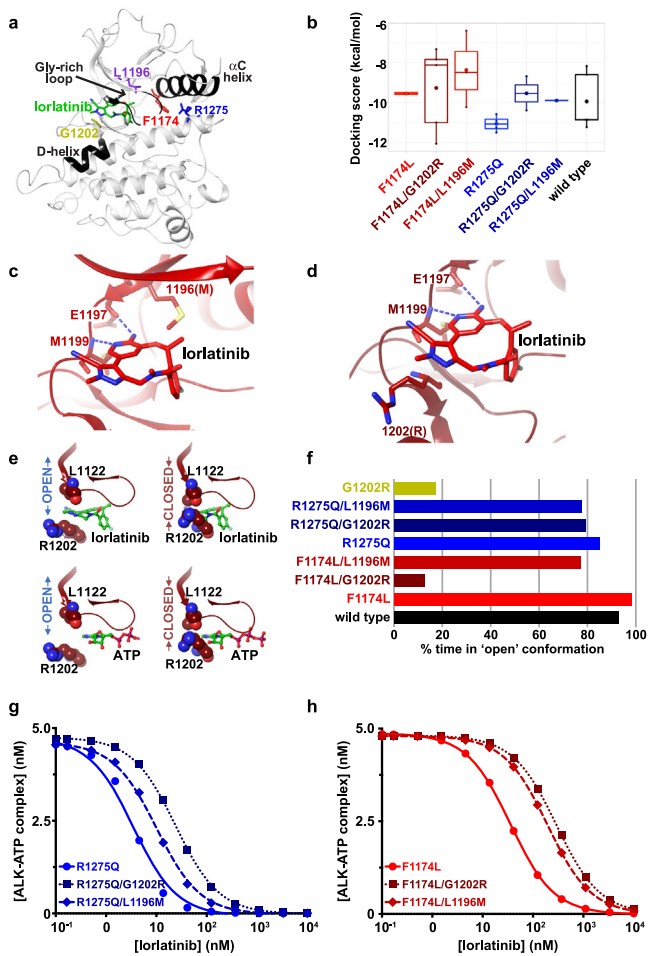

**Fig. 6 | Structural modeling of lorlatinib resistance due to compound mutations.** The color designations for ALK variants in this figure are as follows: F1174L (red) F1174L/G1202R (dark red), F1174L/L1196M (medium red), R1275Q (blue), R1275/G1202R (dark blue), R1275Q/L1196M (medium blue), L1196M (purple), and G1202R (gold). **a** Structure of wild-type ALK-TKD with bound lorlatinib (PDB ID: 4CLI[15]), highlighting the positions of key structural elements, a bound lorlatinib molecule (green), F1174 (red), R1275 (blue), L1196 (purple) and G1202 (gold). **b** Distribution of lorlatinib-docking scores for modeled ALK-TKD variants, with lower (more negative) docking scores corresponding to increased binding energy. The docking scores were generated with $n$ of 10 independently run induced fit docking experiments. For each variant, the filled symbol in the boxplot represents the mean, the horizontal line within the box represents the median, and upper and lower bounds of the box represent the 75th and 25th percentiles (to give interquartile range). The upper and lower whiskers represent the maximum and minimum values of the data that are within 1.5 times the interquartile range. **c** Best-fit docked lorlatinib (medium red) pose in the F1174L/L1196M ALK-TKD variant (docking score −10.213 kcal/mol), with lorlatinib shown in red and conserved interactions (blue dashed lines) with E1197 and M1199 of the hinge region marked, as well as position 1196 (replaced with M). **d** Best-docked lorlatinib pose for F1174L/G1202R (docking score −12.033 kcal/mol) showing conserved interactions (blue dashed lines) with E1197 and M1199 of the hinge region and the arginine introduced at position 1202. **e** Positions of R1202 and L1122 in the F1174L/G1202R ALK-TKD model in both the gate-open (left) and gate-closed (right) conformations, showing that ATP (green) placed as in PDB ID 3BUS[55] (bottom) can bind either conformation, whereas lorlatinib placed as in PDB ID: 4CLI[15] (top) can only bind the gate-open conformation. **f** Plot of the percentage of time in the 300 ns MD simulations that each ALK-TKD variant noted spends in the gate-open conformation. Note that the F1174L/G1202R variant spends 10% of the time with the gate 'open', and ~90% with the gate closed. **g** Lorlatinib inhibition curves generated in silico for R1275Q-based variants as described in Methods and Supplementary Note 2, with [ALK-TKD] set at 5 nM, and [ATP] at 1 mM, using only the gate-open kinetic model. **h** Lorlatinib inhibition curves generated in silico for F1174L-based variants as described in **h**. Curves for ALK-TKD with the F1174L (red circles, solid curve) or F1174L/L1196M (medium red diamonds, dashed curve) mutation were unaffected by whether or not the gate was allowed to close (see Supplementary Data Fig. 8e). Only with the gate closed ~90% of the time as predicted in **f** see Supplementary Data Fig. 8a, b) did the F1174L/G1202R variant (dark red squares, dotted curve) show the increased IC$_{50}$ value plotted here. Source data are provided as a Source Data file.

Supplementary Data Fig. 8a–e), with the experimentally observed ~10-fold increase in IC$_{50}$ for F1174L/G1202R ALK-TKD being best approximated when it spends just 10% of the time in the gate-open conformation (Supplementary Data Fig. 8b).

## Discussion

Unlike many adult malignancies, pediatric tumors harbor relatively few mutations at diagnosis, but often show enrichment of potentially targetable mutations at time of relapse[6,7,9,34]. Tissue biopsy is not always feasible in children, does not allow serial monitoring, and may not capture regional heterogeneity. Serial ctDNA sampling, by contrast, offers the potential to assess underlying tumor heterogeneity and its evolution and to detect rare, stochastic, resistance-conferring genetic alterations that are selected for within a tumor cell population during treatment. We and others recently described the potential utility of serial ctDNA profiling in neuroblastoma patients[33–35]. Here, within a prospective biomarker-driven phase 1 pediatric trial[32], we demonstrate that longitudinal ctDNA biopsies coupled with radiographic evaluation of response provide unique insight for understanding how tumors change over time. These data allow the elucidation of mechanisms of response and resistance to ALK inhibition with lorlatinib in patients with relapsed/refractory *ALK*-mutated neuroblastoma and illustrate the utility of ctDNA analysis to track disease course under the selective pressure of targeted therapy and perhaps ultimately intervene pre-emptively before clinical disease progression.

We observed a correlation between *ALK* VAF in ctDNA samples and clinical-radiographic response in a majority of patients, supporting the utility of tracking *ALK* VAF as a non-invasive clinical tool. However, no correlation was observed in a second group of patients, likely due to spatial and temporal intra-tumor heterogeneity, a

complex tumor ecosystem, and the emergence of off-target resistant clones. In a third group of patients, all with objective anti-tumor responses to lorlatinib—many of which were sustained—*ALK* was not detected and ctDNA was likewise undetectable despite serial sampling. It is unclear whether this is due to any patient- or tumor-specific biology, such as those with a more indolent phenotype and diminished disease burden, or to other technical factors. There seemed to be no clear difference between patients receiving lorlatinib as monotherapy compared with those receiving combination therapy. Our data demonstrate that—when detectable—known tumor driver mutations can provide important clinical information about individual patients. When ctDNA is undetectable, however, conclusions naturally cannot be drawn about response to therapy. Furthermore, estimates of total ctDNA content could not be ascertained for every sample, and for some samples could be derived only with manually assessed MSAF, an algorithm based on the maximum somatic variant allele frequency which may yield less accurate assessment depending on the mutations present. Altogether, these findings highlight the striking variability among patients to be overcome as we deliberate how best to proceed with clinical translation of liquid biopsies for patient care. The FoundationOne Liquid CDx test includes a panel of 324 genes, many of which have no relevance to neuroblastoma or other childhood cancers. Pediatric tumor-specific assays are needed to address the unique genomics, diagnostic challenges, and treatments of childhood solid tumors. To address this, we are developing a more focused and applicable gene panel for childhood solid tumors that will maximize sequencing depth and perhaps shed more light on patients such as those in Group 3 with 'Undetectable *ALK*.'

Analysis of serial ctDNA sampling in our cohort revealed the acquisition of mutations during patient treatment with lorlatinib, and we identified acquisition of both compound *ALK* mutations and mutations in RAS-MAPK pathway components as mechanisms of resistance to lorlatinib in patients with ALK-driven neuroblastoma. As demonstrated in studies of ctDNA in large adult cohorts[41], these mutations initially emerge at very low VAF, likely representing sub-clonal outgrowth of a resistant clone, and correlate with a rising detectable *ALK* VAF. Unfortunately, since all patients had clinical progression at these time points, they were taken off study and no further ctDNA samples were collected to evaluate continued rise of the resistant clone. In two cases (patient 5 and 13), the potential resistant clone is present at very low VAF despite very high *ALK* VAF. Patient 5 harbored concurrent ALK amplification and F1174 mutation, which increased at disease progression. The VAF of the potential resistance mutation is therefore further decreased by the total ALK alleles present in the sample, reflecting amplification of the F1174 mutation. In patient 13, both an increase in baseline *ALK* VAF and the newly acquired G1202R mutation were present at the timepoint prior to progression; it is therefore possible that an additional mechanism of resistance is also acting in this context. These data support a new monitoring strategy that can be leveraged in pediatric oncology clinical trials, with resistance mutations occurring at low VAF heralding changes in clinical response. While our experimental data clearly demonstrate that compound mutations confer decreased pharmacologic response to lorlatinib in multiple contexts, it was not possible to collect samples post-progression to determine whether the VAF of these compound mutations increases with tumor burden. Further and sustained serial sampling in larger clinical trials would be necessary to definitively ascertain the clinical impact of these mutations in children receiving lorlatinib.

Importantly, mutations in the *RAS-MAPK* pathway represented the most common mechanism of off-target genetic resistance to lorlatinib, as also reported in other studies[42]. Patients with concurrent pathogenic *ALK* and *RAS-MAPK* pathway mutations at trial enrollment did not respond to lorlatinib, and acquisition during treatment of new mutations in this pathway heralded disease progression. The prognostic effect of *RAS-MAPK* pathway mutations at diagnosis is currently unknown, but they are enriched at disease relapse. Together, these findings raise the question of the role of RAS pathway inhibitors in potentially preventing disease relapse and resistance to ALK inhibition, which must be addressed experimentally. Such an approach would be of potential benefit to the subset of patients with ALK-driven neuroblastoma who either acquire mutations in the *RAS-MAPK* pathway as the disease circumvents lorlatinib, or may suggest a strategy for patients who present with dual *ALK* and *RAS-MAPK* pathway mutations. These findings further support the utility of liquid biopsies in elucidating mechanisms of polyclonal resistance.

Another key goal of this study was to monitor the evolution of on-target resistance mechanisms through acquisition of *ALK* compound mutations as seen in NSCLC[13]. The progressive accumulation of on-target mutations in adult patients with *ALK* fusion-positive NSCLC receiving successive generations of ALK inhibitor[13,16] has led to the clinical conundrum of which drug to initiate in frontline therapy. In neuroblastoma, where ALK is activated by point mutations in the TKD of the full-length receptor, prior ALK inhibition did not appear to impact the time (or development) of on-target resistance, contrasting with the experience in *ALK* fusion NCLSC[13]—noting that our assessment is limited by the relatively small number of patients studied here. We confirmed that the compound *ALK* mutations identified in patients receiving therapy with lorlatinib also confer resistance in cellular and biochemical models. In *ALK* fusion-driven NSCLC, patients exposed to earlier-generation ALK inhibitors develop single *ALK* mutations, including L1196M and G1202R that can be overcome by lorlatinib[13,16,18,20,21,37], but the L1196M/G1202R compound mutation

causes lorlatinib resistance[13,20]. In lorlatinib-resistant neuroblastoma patients, G1202R and L1196M mutations are instead paired separately with the neuroblastoma-specific hotspot activating mutations F1174L or R1275Q. Importantly, beyond activating ALK in neuroblastoma, F1174 mutations also emerge as an acquired resistance mutations in *ALK* fusion-positive tumors treated with crizotinib[43] or second-generation ALK inhibitors[44]. The F1174L/G1202R (or F1174C/G1202R) combination has also been seen in lorlatinib resistant NSCLC patients[17,21,37], whereas F1245 and R1275 mutations have not been seen in NSCLC in any context.

Although L1196M and G1202R mutations in *cis* with F1174L were acquired in patients on this study, currently none of the 11 patients treated at the RP2D of lorlatinib (115 mg/m²) have yet developed compound resistance mutations. Moreover, we have not seen the R1275Q/L1196M combination in our studies, and are not sure whether the secondary mutations observed with R1275Q and F1245V are in *cis* or in *trans*. It seems possible that increased lorlatinib exposure with the RP2D could overcome the loss of sensitivity seen for the F1174L/L1196M mutant and other compound mutations—with the possible exception of the F1174L/G1202R variant where the gate closure mechanism may be confounding. Other lorlatinib-resistant compound mutations in NSCLC have included G1128A, T1151M, C1156Y, I1171S/T/N, G1269A, and D1203N[13,17,18,21,37], of which G1128A, T1151M, and I1171N have also been seen as low frequency activating mutations in neuroblastoma[10]. Since these mutations partly resemble F1174L in their effects on the properties of the TKD[10] it will be interesting to see if they are more likely than R1275Q to be found alongside G1202R or L1196M in lorlatinib-resistant neuroblastoma cases in which they function as a driver.

In conclusion, the data presented here demonstrate the significance of analyzing liquid biopsies in providing clinically relevant information on disease and treatment course and determining mechanisms of resistance. More comprehensive studies will be necessary to fully determine clonal evolution of genomic alterations under the pressure of lorlatinib therapy. We also speculate that the lorlatinib RP2D established in the first-in-children clinical trial—at about twice the recommended adult dose—may be important in preventing emergence of compound *ALK* mutations. This hypothesis is being addressed formally in the current Children's Oncology Group Phase 3 study for patients with ALK-driven high risk NB (NCT03126916) where lorlatinib has replaced crizotinib during frontline therapy. Prospective serial ctDNA samples will be collected on this study and are expected to yield further insights into clinical response and resistance in therapy naïve patients.

## Methods

### Trial design and patient cohort

Circulating tumor DNA was obtained from patients enrolled on the NANT 1502 Phase I/IIb study of lorlatinib (NCT03107988, clinicaltrials.gov). Each site's institutional review board approved the protocol and consent (Children's Hospital Los Angeles Institutional Review Board, CA, USA; Emory University Institutional Review Board, GA, USA; Children's Hospital of Philadelphia Institutional Review Board, PA, USA; Cincinnati Children's Hospital Institutional Review Board, OH, USA; Colorado Multiple Institutional Review Board, CO, USA; Cook Children's Health Care System Institutional Review Board, TX, USA; Comité de Protection des Personnes Ile-de-France X, Paris, France; Dana-Farber Cancer Institute, Office for Human Research Studies, MA, USA; London City & East Research Ethics Committee, Bristol, UK; Seattle Children's Institutional Review Board, WA, USA; SickKids Research Ethics Board, Toronto, CA; The University of Chicago Biological Sciences Division Institutional Review Board, IL, USA; University of California San Francisco Human Research Protection Program Institutional Review Board; University of Michigan Medical School Institutional Review Board, MI, USA). The trial was conducted in

accordance with the Declaration of Helsinki, International Conference on Harmonization guidelines for Good Clinical Practice and local regulations. Patients or legal guardians provided informed consent and assent was obtained per institutional guidelines.

Forty-nine patients with ALK-mutated relapsed/refractory high risk NBL were enrolled on the NANT 1502 study between September 2017 and February 2022[32]. Twenty-four (49%) of patients were male and 25 (51%) were female. Blood samples for circulating tumor DNA profiling were drawn prior to enrollment, after courses 2, 4, and 6, and then after every 4 courses of treatment for the duration of patient participation in the trial. Forty-six patients were evaluable for this ctDNA study. Clinical data were obtained per study protocol and centrally reviewed as described[32].

## ctDNA sequencing and analysis

16-20 ml of whole blood was drawn from each patient and sent to Foundation Medicine in Streck tubes for ctDNA profiling. From October 2017 to June 2021, samples were profiled using the FoundationACT ligation-adapter and hybrid-capture based NGS assay, a 62 gene panel[45]. From August 2021, samples were profiled using the now FDA-approved FoundationOneLiquid CDx platform, a 324 gene panel[36]. From the 47 evaluable patients in our cohort, 149 samples sent through July 2022 met quality thresholds for data analysis (123 samples profiled with FoundationACT and 26 with FoundationOneLiquid CDx).

Sequencing data were analyzed for the presence of short variants (single base substitutions and short insertions/deletions), copy number amplifications and homozygous deletions, and detectable gene rearrangements. For downstream analyses, we utilized only short variants, amplifications, homozygous deletions, and rearrangements with pathogenic functional significance. Abnormalities with unknown significance (VUS: variants of unknown significance) were discarded.

For most samples, we were able to estimate the ctDNA quantity using the comprehensive tumor fraction (CTF) method, using established computational methods[46,47]. For samples in which no *ALK* was detected, we analyzed CTF to determine if any ctDNA was found in the cell-free DNA sequenced. For samples where CTF could not be calculated, MSAF was manually assessed to determine the fraction of ctDNA content[48].

For the compound ALK mutations, strand assessment was determined by manually analyzing the individual sequencing reads when applicable, and determining whether the mutations occurred in *cis* or in *trans*.

## Cell lines and reagents

Kelly and CHLA-20 neuroblastoma cell lines were obtained from the Children's Hospital of Philadelphia cell bank and routinely genotyped by short tandem repeat (STR) and tested for mycoplasma. Kelly cells were cultured in RPMI 1640 media supplemented with 10% fetal bovine serum (FBS), 1% L-glutamine, and 1% penicillin/streptomycin, maintained at 370 C and 5% $CO_2$. CHLA-20 cells were grown in Iscove's Modified Dulbecco's Medium (IMDM), 20% FBS, 1% L-glutamine, and 1% insulin, transferrin, and selenium (ITS; Sigma Aldrich).

## CRISPR knock-in

The Alt-R CRISPR system (Integrated DNA Technologies) was used to design *ALK* guides (1196 and 1202 sequence). We used gRNA targeting *ATP1A1* G4 (GTTCCTCTTCTGTAGCAGCT), previously reported to improve the efficiency of HDR after selection with ouabain[38]. The crRNA and tracrRNA were resuspended to 100 μM stock solutions in Nuclease-Free IDTE Buffer (Integrated DNA Technologies). Equimolar concentrations of crRNA–tracrRNA complexes of RNA oligonucleotides were mixed, heated at 95°C for 5 min, and cooled to room temperature. To prepare RNP complexes, 150 pmol of gRNA was mixed with 125 pmol of Alt-R Cas9 enzyme (62 μM) and incubated for 20 min. RNP complexes were nucleofected with 400 pM of each ssODN using a

Lonza 4D nucleofector. HDR Enhancer V2 was added to the medium to improve editing efficiency. Cells were incubated at 37°C for 48 h in a 96-well plate, transferred to 12-well plates, and selected with 0.5 μM ouabain for 96 h. Surviving cells were seeded in 96-well plates for single-cell colony selection. Single clones were expanded, and DNA editing was validated by sequencing. Initial screening was performed by extracting DNA from single colonies and sequencing PCR products using mutation-specific reverse primers (for L1196 mutation- CAT GAGCTCCATGAGGATG and for the G1202 mutation- CTTGAGGTCT CTGCCGG), using a forward primer that binds upstream of the F1174 mutation GCTCTGCAGCAAATTCAAC. For screening for the G1202 mutation in CHLA-20, PCR-amplified DNA fragments using ACACTTCCTCACCCAAGTGC and CCATCGAGGAACTTGCTACC primers were sequenced to determine heterozygous incorporation of ALK mutation at position 1202, RNA was extracted from cells, cDNA was made, and PCR amplified using CAAGTGGCTGTGAAGACGCT and CCTTCCATGAAGGCCTCTG primers. *Cis* incorporation of mutations was confirmed using TA cloning and sequencing of single bacterial colonies.

## Cell viability assays

3000 cells per well were plated in tissue culture-treated 96-well plates and were treated with DMSO or the noted concentration of lorlatinib 24 h later. Cell viability was assessed 120 h after treatment using Cell Titer-Glo luminescent cell viability assay (Promega) with a GloMax Reader (Promega). Graphpad Prism 9 was used to plot for viability curves and calculate $IC_{50}$ values for lorlatinib.

## ALK recombinant protein expression and purification

DNA encoding ALK residues 1090-1416 (in precursor protein numbering), together with an N-terminal hexahistidine tag, was subcloned into pFastBac-1 (Invitrogen) for expression of histidine-tagged recombinant ALK TKDs. Constructs for expressing compound ALK variants were derived using the Q5 site directed mutagenesis kit (New England BioLabs) using CATGGCGGGGAGAGACCTCAA and AGCTC CAGCAGGATGAAC primers for G1202R and GTTCATCCTGATG GAGCTCATGGC and CGGGGCAGGGATTGCAGG for L1196M in the background of ALK F1174L or R1275Q plasmids used in reference 22. ALK-TKDs were produced in *Spodoptera frugiperda* Sf9 cells (ATCC− CRL1711) cells using recombinant baculovirus with the Bac-to-Bac expression system (Invitrogen). Sf9 cells were infected with recombinant baculovirus for 3 days at 27°C and harvested by centrifugation at $2250 \times g$ for 15 min. Cell pellets were washed with phosphate-buffered saline, resuspended in lysis buffer [20 mM Tris-HCl, pH 8 at 4°C, 300 mM NaCl, 5 mM 2-mercaptoethanol, 10 mM imidazole, 5% glycerol, plus cOmplete EDTA-free protease inhibitor cocktail (Roche)], lysed by sonication and cleared by centrifugation for 1 h at $74\,766 \times g$. Cleared cell lysate was applied to a 1.5 ml bead bed of Ni-NTA agarose (Qiagen), which was then washed twice with 20 volumes of 'Wash-1' [lysis buffer + 200 mM NaCl] and 'Wash-2' [lysis buffer + 15 mM imidazole]. Protein was then eluted with 5 ml of lysis buffer containing 250 mM imidazole. The eluate was filtered (0.22 μm) and applied to a HiLoad 16/60 Superdex 200 column (Cytiva) equilibrated in 25 mM HEPES pH 7.5, 150 mM NaCl, with 100 μM tris(2-carboxyethyl)phosphine (TCEP). Peak fractions were pooled and YopH phosphatase added at 1 μM for 15−20 min at room temperature to reverse spontaneous phosphorylation of ALK-TKD that may have occurred during production. To separate dephosphorylated ALK from residual phosphorylated protein, and to remove YopH, the protein was then applied to a 1 ml HiTrap Q HP anion exchange column (Cytiva) after 10-fold dilution into buffer A [20 mM HEPES, pH 7.5, with 100 μM TCEP] and eluted with buffer B [20 mM HEPES, pH 7.5, 1 M NaCl, with 100 μM TCEP] using step elution at 10% B for 5 column volumes followed by gradient elution (from 10%−40% B over 20 column volumes). Dephosphorylated ALK-TKD eluted at 10.8 mS/cm (-120 mM NaCl).

Dephosphorylation was confirmed by intact mass spectrometry. Protein concentrations were determined by absorbance at 280 nm using a calculated extinction coefficient of 39880 $M^{-1} cm^{-1}$, and purity was checked by SDS-PAGE imaged using a Bio-Rad GelDoc-EZ imager, running Image Lab Version 5.2.1. Proteins were used immediately for kinase and $IC_{50}$ assays.

## In vitro kinase assays

Kinase assays were conducted with PhosphoSense® Sox-based fluorophore technology (AssayQuant) using ALK sensor AQT0101 as peptide substrate, which contains a sulfonamido-oxine (Sox) fluorophore that shows chelation-enhanced fluorescence upon peptide phosphorylation[49,50]. Reaction conditions consisted of 50 mM HEPES (pH 7.5), 0.012% Brij-35, 10 mM $MgCl_2$, 1% w/v glycerol, 0.1 mg/ml bovine serum albumin (BSA), and 1 mM dithiothreitol (DTT). For $K_{M,peptide}$ determination, sensor substrate peptide concentrations were varied from 0–250 µM and ATP was kept at a saturating concentration (1 mM). For $K_{M,ATP}$ determination, ATP concentration was varied from 0–2 mM and peptide substrate kept at 10 µM. Reactions of 20 µl total volume were conducted in a 384-well assay plate format at 30°C and initiated with addition of kinase after 15 min of pre-incubation at 30°C. Kinases were diluted in enzyme dilution buffer [20 mM HEPES, pH 7.5, 0.01% Brij-35, 5% v/v glycerol, 1 mM DTT, 1 mg/ml BSA]. The final reaction concentration for all ALK-TKDs was 5 nM, chosen so that reaction rate is linear with enzyme concentration. Phosphorylation progress curves were monitored using a BioTek Synergy microplate plate reader with the fluorescence intensity module set at 360 nm/480 nm excitation/emission wavelengths. Reads were taken every 2.06 min for 3–4 h. Background fluorescence was determined with a "no kinase" control and subtracted from the total signal to obtain corrected RFU. To convert corrected RFU/min to µM peptide substrate/min, reactions were conducted with varying concentrations of peptide substrate 0–250 µM and 1 µM ALK F1174L TKD to achieve maximal phosphorylation. Fluorescence intensity counts were plotted against known concentrations of peptide substrate to generate a standard curve. Initial rates (determined at <20% product conversion) were calculated by selecting the linear portion of reaction progress curves (typically within the first 12 min). $K_M$ values were calculated by plotting the reaction rate against peptide substrate concentration (for $K_{M,peptide}$) or ATP concentration (for $K_{M,ATP}$) and fitting to the Michaelis–Menten equation ($v_o = V_{max}[S]/(K_M + [S])$) using GraphPad Prism 9.2. Values are reported as mean ± standard deviation (SD) of at least three biological replicates.

## IC$_{50}$ determination

The half maximal inhibitory concentrations ($IC_{50}$) of lorlatinib (SelleckChem PF-6463922 Cat. No. S7536) and crizotinib (SelleckChem PF-02341066 Cat. No. S1068) for ALK-TKD variants were determined using the assay conditions above with varying concentrations of inhibitor (0-10000 nM), with fixed 1 mM ATP, 10 µM peptide substrate and 5 nM kinase. Inhibitor dilutions were made in 100% DMSO, further diluted 1:10 with 1× kinase buffer [50 mM HEPES, pH 7.5, 0.01% Brij-35, 10 mM $MgCl_2$] and 2 µl was added per reaction so that final DMSO concentration was constant at 1%. Initial rates were determined as described under the kinase assay method. Curves were normalized against the maximal velocity ("no inhibitor" condition). $IC_{50}$ values were determined by fitting a 3-parameter fit [Inhibitor] vs normalized response using GraphPad Prism 9.2. $IC_{50}$ values are reported as mean ± SD for three biological replicates.

## Focus formation assays

Low-passage NIH 3T3 cells were cotransfected with the MigR1 vector[51] (engineered to express each ALK variant) and pSVneo DNA in a 10:1 ratio using Lipofectamine 2000 (Invitrogen). Cells were allowed to recover for 2 days post-transfection, and then dilutions were split across two six-well plates. The first set of samples were allowed to reach confluence and form foci in DMEM containing 5% calf serum. The second set of samples were selected for colony formation in DMEM containing 10% calf serum and 0.5 mg/ml G418. After 10–14 days, cells were fixed in 3.7% formaldehyde in PBS for 5 min and stained with 0.05% crystal violet. Foci were counted and normalized by the number of G418-resistant colonies for the same variant and plotted as a percentage of the foci formed with F1174L ALK. Three biological repeats were performed in duplicate and plotted (Fig. 5f) as mean ± SD.

## Modeling methods

A model for the inactive wild-type ALK-TKD was generated using MODELLER v. 9.24[52], adding C-terminal residues 1084–1095 and N-terminal residues 1400–1405 from PDB entry 4FNW[53] [https://www.rcsb.org/structure/4FNW] to the structure in PDB entry 3LCS[54] [https://www.rcsb.org/structure/3LCS]. All ligands were removed. Mutations were introduced into this inactive wild-type ALK-TKD structural model using BioPhysCode Automacs script based on MODELLER (https://biophyscode.github.io/). All files can be found on (https://github.com/witekgabriela/StructuralStudy_ALKCompoundMutations).

The ALK-ATP complex was modeled based on the insulin receptor TKD complex with $Mg^{2+}$-ATP from PDB entry 3BU5[55] [https://www.rcsb.org/structure/3BU5]. The inactive ALK-TKD model was aligned to 3BU5 structures using residues R1253-C1255 and G1269-F1271 to position the $Mg^{2+}$ ion near the conserved ALK N1254 and D1270 side chains in the catalytic loop that chelate $Mg^{2+}$. Lastly, nine water molecules that showed optimal ATP binding were copied from PDB ID 3BU5 (see Molecular docking section).

## Molecular dynamics (MD)

All structures were subjected to the same molecular dynamics (MD) protocol. MD was run using GROMACS 2018 software[56]. Each system topology file was generated using the CHARMM27 forcefield for protein[57] and TIP3P forcefield for water[58]. The solvation box was first set around the center of the protein, and the protein placed 1.2 nm from the edge of the triclinic periodic box. Each system was then subjected to several energy minimization and relaxation steps before the main MD run. In the first step, each system was solvated with 0.15 M NaCl, resulting in zero system charge. The energy of the system was next minimized to remove steric clashes or incorrect geometries, using the step-descent minimization algorithm over 0.5 ns. In the following two steps, solvent was equilibrated around the restrained protein so that the temperature and then pressure reached set values. Solvent equilibration was done with temperature coupling using a modified Berendsen thermostat[59] and protein restraint at 300 K. Equilibration was continued with pressure coupling using the Parrinello-Rahman barostat[60] over 100 ps to reach 1 bar. Lastly, the main production MD was run for a total of 400 ns at 300 K and 1 bar with restraints removed. The first 100 ns was ignored in the analysis, and 100 ns sections of the subsequent 300 ns were selected as ($n = 3$) replicates. All simulations were run using XSEDE resources[61]. Analysis of MD ($n = 3$) for each ALK variant was used to calculate the time ALK is present in gate closed or gate open binding pocket conformations.

## Molecular docking

All structures were subjected to the same molecular docking protocol. Induced fit docking (IFD) was performed using the Glide[62–64] and Prime[65,66] module developed by Schrödinger, run using Maestro software version 2019-4 (Schrödinger Release 2019-4: Maestro, Schrödinger, LLC, New York, NY, 2021). IFD was performed on ten structures of each ALK variant generated every 10 ns, between 100–200 ns. Molecule 5P8 in crystal structure 4CLI[15] [https://www.rcsb.org/structure/4cli] was prepared as a template for lorlatinib. Hydrogens were added to structure 5P8 to match a pH of 7, and the chirality was maintained, keeping the lowest energy conformation. Using the ALK

crystal structure in complex with lorlatinib (PDB: 4CLI[15]) as a reference, water molecules were introduced into the binding. Docking of lorlatinib to the reference structure with- and without water molecules showed that the presence of water molecules allowed the docking method to replicate the lorlatinib configuration seen in the wild-type ALK-TKD crystal structure with an improved docking score (−10.263 kcal/mol vs. −8.374 kcal/mol) (Supplementary Data Fig. 9a, b). IFD was run using the standard protocol[67,68], generating up to 20 poses, with ligand conformational sampling energy window for ring conformation set to 2.5 kcal/mol, and receptor and ligand van der Waals scaling set to 0.5. Residue refinement using Prime was set to 5 Å, and subsequent redocking was done using the SP method and included structures within 30 kcal/mol of the previously best-docked[69]. The box center was established using residues 1198 and 1203. The size of the box was approximated by the ligand size. Minimization of the protein during IFD was performed using the OPLS3e forcefield[70]. Docking of ATP was performed using the IFD protocol with the inactive ALK-TKD structure containing one $Mg^{2+}$ ion and nine water molecules. The box center was established using residues 1270, 1201 and 1150, and box size was set to 15 Å.

### Modeling of ALK inhibition by lorlatinib

A computational kinetic model of lorlatinib competitive binding to ALK-TKD in the presence of ATP was used to model resistance arising from either a decrease in lorlatinib binding energy or the presence of a closed (occluded) drug binding. We modeled $IC_{50}$ for lorlatinib as the drug concentration at which 50% of the ALK/ATP complex is depleted as lorlatinib outcompetes ATP for the binding site—recapitulating the biochemical lorlatinib $IC_{50}$ described in this study. The kinetic model described in Supplementary Data Fig. 9c was used, and the modeled $IC_{50}$ values were derived using the differential equation tool, COPASI v. 4.27[71] as described in Supplementary Note 2. All $IC_{50}$ values were derived at steady state, as a comparison of stochastic and deterministic models revealed no differences.

### Statistics and reproducibility

Plasma samples were drawn at pre-set time points from patients enrolled in the clinical trial. This Phase I trial was designed as a 3 + 3 dose escalation design in each of the three cohorts[32]. All plasma samples from eligible patients that were drawn appropriately, yielded sufficient DNA for sequencing, and with results that passed Foundation Medicine QC metrics, were utilized for analysis.

For biochemical assays (Fig. 5), no data were excluded from the analysis and no statistical method was used to predetermine sample size. The experiments were not randomized. The investigators were not blinded to allocation during experiments and outcome assessment. The experimental data shown represent the outcomes of three independent biological experiments, each performed in technical triplicate. A two-sided Student's *t* test was used to assess differences between means. A *P* value of 0.05 or less was considered statistically significant. In silico statistics and reproducibility information are included in respective methods sections and Supplementary Information.

### Data availability

Materials and correspondence requests should be addressed to Yaël P. Mossé (mosse@chop.edu) or Mark A. Lemmon (mark.lemmon@yale.edu). All COPASI files used to derive $IC_{50}$ and structural PDB files can be accessed via GitHub https://github.com/witekgabriela/StructuralStudy_ALKCompoundMutations [https://doi.org/10.5281/zenodo.7753184] https://zenodo.org/record/7753184#.ZByKrOzMIik. The structural publicly available data used in this study are available in the PDB database under accession code 4CLI, 3LCS, 4FNW, 3BU5. The remaining data are available within the Article, Supplementary Information, or Source Data file. Source data are provided in this paper.

### Code availability

Molecular dynamics code deposited in Github https://github.com/witekgabriela/StructuralStudy_ALKCompoundMutations.

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

## Acknowledgements
We thank the patients and all investigators involved in this study. Lorlatinib was provided by Pfizer, Inc. Contributions to this work by Y.P.M.'s group were supported by National Cancer Institute grants P01 CA217959 and R01 CA140198, Cookies for Kids Cancer, St. Baldrick's Foundation, V Foundation for Cancer Research, Solving Kids Cancer US/UK, Children's Neuroblastoma Cancer Foundation, the Band of Parents, the EVAN Foundation, Wade's Army, Ronan Thompson Foundation, and the Catherine Elizabeth Blair Memorial Foundation. Also supported by a Conquer Cancer Young Investigator Award (E.R.B.), by Doris Duke Foundation (E.R.B.), by NCI grant R35 CA220500 (J.M.M.), by TL1 TR001880-03 (G.M.W.), and by NIGMS grant R35 GM122485 (M.A.L.).

## Author contributions
Y.P.M. conceived the overall project. E.R.B. performed all patient data analysis, assisted by D.P., A.D., and M.G. G.W. performed all molecular modeling and studies together with I.G., supervised by R.R., S.M., G.W., J.K., A.K., C.C., and K.K. performed the work on modeling compound mutations in neuroblastoma cell lines, supervised by Y.P.M. Z.O.P., M.A.W., and C.M.S. performed all biochemical studies, supervised by M.A.L., Y.P.M., M.A.L., and R.R. supervised the overall project. E.R.B., M.A.L., and Y.P.M. drafted the manuscript, and all authors were involved in data interpretation, writing, and critical review of the manuscript. All authors have approved the submitted version and are accountable for their contributions and the integrity of the work.

## Competing interests
Y.P.M. is the Principal Investigator of the NANT Phase 1 trial of lorlatinib and is a consultant for Pfizer. No research funding from Pfizer was received for this study. Y.P.M. has previously received research funding from Pfizer and Novartis. Y.P.M. has also served as a consultant for Lilly, Auron Therapeutics, and Jumo Health. Y.P.M. serves as a member of the Data and Safety Monitoring Committee for the ASCO TAPUR study and receives honoraria for this role. D.P. is an employee of Foundation Medicine. The remaining authors declare no competing interests.

## Additional information

[1]Division of Oncology and Center for Childhood Cancer Research, Children's Hospital of Philadelphia, Philadelphia, PA, USA. [2]Division of Pediatric Hematology and Oncology, Schneider Children's Medical Center, Petach Tikva, Israel, Faculty of Medicine, Tel Aviv University, Tel Aviv, Israel. [3]Perelman School of Medicine at the University of Pennsylvania, Philadelphia, PA, USA. [4]Department of Bioengineering, University of Pennsylvania, Philadelphia, PA, USA. [5]Department of Pharmacology, Yale University School of Medicine, New Haven, CT, USA. [6]Yale Cancer Biology Institute, Yale University, West Haven, CT, USA. [7]Foundation Medicine, Inc, Cambridge, MA, USA. [8]Cancer and Blood Disease Institute, Children's Hospital Los Angeles, Los Angeles, CA, USA. [9]Keck School of Medicine, University of Southern California, Los Angeles, CA, USA. [10]St. Jude Children's Research Hospital, Memphis, TN, USA. [11]Aflac Cancer and Blood Disorders Center, Children's Healthcare of Atlanta, Atlanta, GA, USA. [12]Winship Cancer Institute, Emory University School of Medicine, Atlanta, GA, USA. [13]Seattle Children's Hospital, Seattle, WA, USA. [14]Department of Chemical and Biomolecular Engineering, University of Pennsylvania, Philadelphia, PA, USA. [15]These authors contributed equally: Esther R. Berko, Gabriela M. Witek, Smita Matkar, Zaritza O. Petrova. ✉e-mail: mark.lemmon@yale.edu; mosse@chop.edu

