## [Peer Review File · Nature Communications]

Reviewers' Comments:

Reviewer #1:

Remarks to the Author:

This revision from Berko et al is a significant improvement over the initially submitted manuscript. In particular, the addition of Ext Data Fig 3 addresses one of my major comments from my prior round of review. It remains noteworthy that, in cases where the ctDNA content is being estimated by the VAF of an SNV, especially one that is not known to be disease initiating, this VAF may not accurately reflect total ctDNA content and a comment acknowledging this limitation should be included.

However, I remain concerned about the conclusion that new ALK resistance mutations are contributing to treatment resistance. While the authors have added a statement in the results that these variants were discovered "significantly prior to progression" that seems to be directly contradictory to what is indicated by the red dots in Figure 4 and the statement in the discussion that says "since all patients had clinical progression at these timepoints, they were taken off study and no further ctDNA samples were collected". If these progressions really did happen significantly after the blood samples were drawn, the authors need to show the time between blood draw and clinical progression. Was it days, weeks, months? Figure 4 should be amended accordingly. Furthermore, Figure 4 also clearly demonstrates that VAF (and presumably total ctDNA) is rising at or before the time of the last sample drawn after significant treatment response in VAF levels. Yet, in all cases, this ctDNA-level progression is dominated by the original ALK variant and the new variant is present in only a very small minority of the ctDNA being shed at that time. Therefore, there is clearly a level of tumor progression (whether clinical or subclinical) that is being primarily driven by a mechanism other than these secondary ALK mutations. This remains a confusing part of the manuscript and requires further edits before the manuscript is acceptable for publication.

I am satisfied with the authors' responses to major comments 3 and 4 and minor comments 1-6 from my initial review.

While I agree with the authors that pediatric-specific assays are likely needed, my prior comment #7 was intended to point out that it remains extremely unlikely that one assay will be sufficient to serve all purposes (early cancer detection, accurate diagnosis, precision medicine selection, detection of MRD, study of tumor evolution and treatment resistance, and early detection of relapse) for all pediatric cancers driven by different classes of variants (NB, osteosarcoma, Ewing, brain tumors, etc). Perhaps the simplest solution to addressing this concern is to pluralize the statement to say that pediatric "assays" are likely needed and a comment where the ongoing work by the authors to develop a new assay would fit into this need.

Reviewer #2:

Remarks to the Author:

The authors have adequately addressed my concerns.

Reviewer #3:

Remarks to the Author:

The authors have significantly improved the original manuscript that had been submitted to Nature

Medicine. The vast majority of my concerns have been addressed appropriately. Unfortunately, however, the authors did not address my central point of criticism, which was on very low allele fractions of compound mutations. While I agree that these mutations might have been captured at an early phase of relapse/progression, and that it is conceivable that they confer lorlatinib resistance at clinical progression, this statement remains speculation without providing evidence that such mutations grow out to significant levels over the course of disease. I consider it essential to demonstrate significant increase of such mutations in at least one or two patients, either in ctDNA or in tumor samples, to validate the biological concept and the clinical relevance of compound ALK mutations. It has to be noted that it is also conceivable that VAF of such mutations do not increase at clinical progression, or may even disappear, which would clearly argue against the clinical significance of this finding. Convincing data on this issue is extremely important, as it may guide physicians' decision making on ALK inhibitor treatment in the future, but may lead to inappropriate decisions if not valid.

Reviewer #1 (Remarks to the Author):

1. This revision from Berko et al is a significant improvement over the initially submitted manuscript. In particular, the addition of Ext Data Fig 3 addresses one of my major comments from my prior round of review. It remains noteworthy that, in cases where the ctDNA content is being estimated by the VAF of an SNV, especially one that is not known to be disease initiating, this VAF may not accurately reflect total ctDNA content and a comment acknowledging this limitation should be included.

We thank the reviewer for this feedback and have added the following statement to the discussion (Lines 393-396):

“Furthermore, estimates of total ctDNA content could not be ascertained for every sample, and for some samples could be derived only with manually assessed MSAF, an algorithm based on the maximum somatic variant allele frequency which may yield less accurate assessment depending on the mutations present.”

However, I remain concerned about the conclusion that new ALK resistance mutations are contributing to treatment resistance. While the authors have added a statement in the results that these variants were discovered “significantly prior to progression” that seems to be directly contradictory to what is indicated by the red dots in Figure 4 and the statement in the discussion that says “since all patients had clinical progression at these timepoints, they were taken off study and no further ctDNA samples were collected”. If these progressions really did happen significantly after the blood samples were drawn, the authors need to show the time between blood draw and clinical progression. Was it days, weeks, months? Figure 4 should be amended accordingly. Furthermore, Figure 4 also clearly demonstrates that VAF (and presumably total ctDNA) is rising at or before the time of the last sample drawn after significant treatment response in VAF levels. Yet, in all cases, this ctDNA-level progression is dominated by the original ALK variant and the new variant is present in only a very small minority of the ctDNA being shed at that time. Therefore, there is clearly a level of tumor progression (whether clinical or subclinical) that is being primarily driven by a mechanism other than these secondary ALK mutations. This remains a confusing part of the manuscript and requires further edits before the manuscript is acceptable for publication.

The Reviewer is quite correct in their comment regarding the phrase “significantly prior to progression”, and we appreciate them pointing it out. Only in patient 13 was the compound mutation detected at a timepoint prior to progression. Otherwise, the variants were present *at* disease progression, not prior.

We have removed the phrase “significantly prior to progression” and edited and made sure that this is clear in the results section (Lines 415-417) and the legend to figure 4. We thank the reviewer for highlighting this ambiguous statement, apologize for the terminology, and appreciate the opportunity to rectify this.

The Reviewer comments on the low VAF values, specifically in the context of the dominating ALK variant. As we mention above, this is to be expected. Extensive literature, referenced below¹⁻¹⁶, has demonstrated the clinical impact and relevance of low VAF resistance clones. To further clarify the point about low VAFs, we have added discussion of the specific 2 cases with very high baseline ALK VAF. Patient 5 harbored concurrent ALK amplification and an F1174 mutation. VAF of this mutation increased with disease progression, and a low frequency resistance SNV could have been further diluted by the presence of multiple copies of non-mutated ALK in the sample. Patient 13 likewise demonstrated very high ALK VAF that was elevated with the presence of the compound mutation at the timepoint prior to clinical progression. We therefore highlighted the possibility of alternative mechanisms of resistance and further modified the text to address the Reviewer’s concern, as described in the text below (Lines 411-419):

“In two cases (patient 5 and 13), the potential resistant clone is present at very low VAF despite very high ALK VAF. Patient 5 harbored concurrent ALK amplification and F1174 mutation, which increased at disease progression. The VAF of the potential resistance mutation is therefore further decreased by the total ALK alleles present in the sample, reflecting amplification of the F1174 mutation. In patient 13, both an increase in baseline ALK VAF and the newly acquired G1202R mutation were present at the timepoint prior to progression; it is therefore possible that an additional mechanism of resistance is also acting in this context. These data support a new monitoring strategy that can be leveraged in pediatric oncology clinical trials, with resistance mutations occurring at low VAF heralding changes in clinical response.”

2. I am satisfied with the authors’ responses to major comments 3 and 4 and minor comments 1-6 from my initial review.

We thank the Reviewer.

3. While I agree with the authors that pediatric-specific assays are likely needed, my prior comment #7 was intended to point out that it remains extremely unlikely that one assay will be sufficient to serve all purposes (early cancer detection, accurate diagnosis, precision medicine selection, detection of MRD, study of tumor evolution and treatment resistance, and early detection of relapse) for all pediatric cancers driven by different classes of variants (NB, osteosarcoma, Ewing, brain tumors, etc). Perhaps the simplest solution to addressing this concern is to pluralize the statement to say that pediatric “assays” are likely needed and a comment where the ongoing work by the authors to develop a new assay would fit into this need.

We fully agree and have made this clarification to the text in the following statement (Lines 400-401):
“Pediatric tumor specific assays are needed to address the unique genomics, diagnostic challenges, and treatments of childhood solid tumors. “

Reviewer #2 (Remarks to the Author):

The authors have adequately addressed my concerns.

We thank the reviewer for this feedback.

Reviewer #3 (Remarks to the Author):

1. The authors have significantly improved the original manuscript that had been submitted to Nature Medicine. The vast majority of my concerns have been addressed appropriately.

We thank the Reviewer for this feedback.

2. Unfortunately, however, the authors did not address my central point of criticism, which was on very low allele fractions of compound mutations. While I agree that these mutations might have been captured at an early phase of relapse/progression, and that it is conceivable that they confer lorlatinib resistance at clinical progression, this statement remains speculation without providing evidence that such mutations grow out to significant levels over the course of disease. I consider it essential to demonstrate significant increase of such mutations in at least one or two patients, either in ctDNA or in tumor samples, to validate the biological concept and the clinical relevance of compound ALK mutations. It has to be noted that it is also conceivable that VAF of such mutations do not increase at clinical progression, or may even disappear, which would clearly argue against the clinical significance of this finding. Convincing data on this issue is extremely important, as it may guide physicians' decision making on ALK inhibitor treatment in the future but may lead to inappropriate decisions if not valid.

We certainly agree with the reviewer's comment that subsequent samples showing increasing VAF of these resistance mutations would strengthen the hypothesis. Unfortunately, however, these samples do not exist. The patients were removed from the study at disease progression, and it is both unethical and infeasible to continue to collect research samples from patients who are no longer enrolled on a trial protocol. Furthermore, these patients all went on to various treatments with salvage therapy, which would have made comparing further clinical and genetic progression impossible in any case.

*The Reviewer's concern was precisely our motivation for going to great lengths to demonstrate that the new compound ALK mutations that we detected do all cause lorlatinib resistance. To do so unequivocally, we used three distinct and orthogonal approaches. Our data show that cells that acquire compound mutations in the ALK TKD have decreased response to lorlatinib, and so are the kinase domains themselves that harbor these mutations less sensitive to lorlatinib. Importantly, **every single patient who developed such a compound mutation had disease progression on lorlatinib**. It therefore seems very difficult to escape the conclusion that these mutations at least contribute to lorlatinib resistance and hope you will agree.*

We nonetheless appreciate the Reviewer's point that we do not have data that show that the VAF of these mutations increases with further clinical progression. As suggested, we have added further clarification and explicitly state this limitation in the results, as below (Lines 419-423):

"While our experimental data clearly demonstrate that compound mutations confer decreased pharmacologic response to lorlatinib in multiple contexts, it was not possible to collect samples post-progression to determine whether the VAF of these compound mutations increases with tumor burden. Further and sustained serial sampling in larger clinical trials would be necessary to definitively ascertain the clinical impact of these mutations in children receiving lorlatinib."

It is well established in the field of cancer that low VAF mutations confer resistance; these data have largely come from tumor tissue, and we are validating this by sampling plasma(1-16). The biology of tumor shedding in plasma remains an open field of study. Investigators have shown that the frequency of

resistant cells even at the site of a tumor is one in just 1000 or 10000(17). So, even if we sample DNA directly from a tumor, the signal (VAF) of the compound mutation is expected to be 1000-10000-fold less than for the driver mutation. In the case of our ctDNA analysis, we have another round of dilution – because ctDNA is taken from circulation, resulting dilution coming from all the cells outside of the tumor. Given this two-stage dilution, the VAF signal is likely to be much weaker than 1:10000. We believe that this is the explanation for the low VAFs in our study – and indeed that this is precisely what is expected. Moreover, our mechanistic data, using three orthogonal methods (including novel CRISPR knock-in assays), all show that that when cells accumulate these compound mutations in cis, resistance to lorlatinib is seen. Further, these mutations were only ever seen in patients who developed resistance to lorlatinib, and never in other patients.

All the patients we report who accumulated compound mutations did so at time of progression. We agree with Reviewer 3 that it would be helpful to have subsequent samples from these patients showing that the VAF rises in the emerging resistant clones, this is simply not possible as these patients were not only removed from protocol therapy but were started on different therapies due to the presence of clinical and radiographic disease progression.

This work represents the largest singular prospective collection of circulating tumor DNA samples in a pediatric oncology cohort treated on targeted therapy and identifies compound mutations in ALK (seen exclusively with lorlatinib resistance) that our in vitro studies unequivocally demonstrate to cause lorlatinib resistance. We hope you will agree that this is an important study that has significant clinical significance, and that it is important to share these data with the broader community.

References

1. Abbosh C, Birkbak NJ, Wilson GA, Jamal-Hanjani M, Constantin T, Salari R, Le Quesne J, Moore DA, Veeriah S, Rosenthal R, Marafioti T, Kirkizlar E, Watkins TBK, McGranahan N, Ward S, Martinson L, Riley J, Fraioli F, Al Bakir M, Gronroos E, Zambrana F, Endozo R, Bi WL, Fennessy FM, Sponer N, Johnson D, Laycock J, Shafi S, Czyzewska-Khan J, Rowan A, Chambers T, Matthews N, Turajlic S, Hiley C, Lee SM, Forster MD, Ahmad T, Falzon M, Borg E, Lawrence D, Hayward M, Kolvekar S, Panagiotopoulos N, Janes SM, Thakrar R, Ahmed A, Blackhall F, Summers Y, Hafez D, Naik A, Ganguly A, Kareht S, Shah R, Joseph L, Marie Quinn A, Crosbie PA, Naidu B, Middleton G, Langman G, Trotter S, Nicolson M, Remmen H, Kerr K, Chetty M, Gomersall L, Fennell DA, Nakas A, Rathinam S, Anand G, Khan S, Russell P, Ezhil V, Ismail B, Irvin-Sellers M, Prakash V, Lester JF, Kornaszewska M, Attanoos R, Adams H, Davies H, Oukrif D, Akarca AU, Hartley JA, Lowe HL, Lock S, Iles N, Bell H, Ngai Y, Elgar G, Szallasi Z, Schwarz RF, Herrero J, Stewart A, Quezada SA, Peggs KS, Van Loo P, Dive C, Lin CJ, Rabinowitz M, Aerts H, Hackshaw A, Shaw JA, Zimmermann BG, consortium TR, consortium P, Swanton C. Phylogenetic ctDNA analysis depicts early-stage lung cancer evolution. *Nature* 2017;**545(7655)**:446-51. doi: 10.1038/nature22364. PubMed PMID: 28445469; PMCID: PMC5812436.
2. Barz MJ, Hof J, Groeneveld-Krentz S, Loh JW, Szymansky A, Astrahantseff K, von Stackelberg A, Khiabani H, Ferrando AA, Eckert C, Kirschner-Schwabe R. Subclonal NT5C2 mutations are associated with poor outcomes after relapse of pediatric acute lymphoblastic leukemia. *Blood* 2020;**135(12)**:921-33. doi: 10.1182/blood.2019002499. PubMed PMID: 31971569; PMCID: PMC7218751.
3. Dentre SC, Leshchiner I, Haase K, Tarabichi M, Wintersinger J, Deshwar AG, Yu K, Rubanova Y, Macintyre G, Demeulemeester J, Vazquez-Garcia I, Kleinheinz K, Livitz DG, Malikic S, Donmez N, Sengupta S, Anur P, Jolly C, Cmero M, Rosebrock D, Schumacher SE, Fan Y, Fittall M, Drews RM, Yao X, Watkins TBK, Lee J, Schlesner M, Zhu H, Adams DJ, McGranahan N, Swanton C, Getz G, Boutros PC, Imielinski M, Beroukhi R, Sahinalp SC, Ji Y, Peifer M, Martincorena I, Markowitz F, Mustonen V, Yuan K, Gerstung M, Spellman PT, Wang W, Morris QD, Wedge DC, Van Loo P, Evolution P, Heterogeneity Working G, the PC. Characterizing genetic intra-tumor heterogeneity across 2,658 human cancer genomes. *Cell* 2021;**184(8)**:2239-54 e39. Epub 20210407. doi: 10.1016/j.cell.2021.03.009. PubMed PMID: 33831375; PMCID: PMC8054914.
4. Dietz S, Christopoulos P, Yuan Z, Angeles AK, Gu L, Volckmar AL, Ogronnik SJ, Janke F, Fratte CD, Zemojtel T, Schneider MA, Kazdal D, Endris V, Meister M, Muley T, Cecchin E, Reck M, Schlesner M, Thomas M, Stenzinger A, Sultmann H. Longitudinal therapy monitoring of ALK-positive lung cancer by combined copy number and targeted mutation profiling of cell-free DNA. *EBioMedicine* 2020;**62**:103103. Epub 20201109. doi: 10.1016/j.ebiom.2020.103103. PubMed PMID: 33161228; PMCID: PMC7670098.
5. Fribbens C, Garcia Murillas I, Beaney M, Hrebien S, O'Leary B, Kilburn L, Howarth K, Epstein M, Green E, Rosenfeld N, Ring A, Johnston S, Turner N. Tracking evolution of aromatase inhibitor resistance with circulating tumour DNA analysis in metastatic breast cancer. *Ann Oncol.* 2018;**29(1)**:145-53. doi: 10.1093/annonc/mdx483. PubMed PMID: 29045530; PMCID: PMC6264798.
6. Hsiehchen D, Bucheit L, Yang D, Beg MS, Lim M, Lee SS, Kasi PM, Kaseb AO, Zhu H. Genetic features and therapeutic relevance of emergent circulating tumor DNA alterations in refractory non-colorectal gastrointestinal cancers. *Nat Commun.* 2022;**13(1)**:7477. Epub 20221203. doi: 10.1038/s41467-022-35144-1. PubMed PMID: 36463294; PMCID: PMC9719461.
7. Jacobs MT, Mohindra NA, Shantzer L, Chen IL, Phull H, Mitchell W, Raymond VM, Banks KC, Nagy RJ, Lanman RB, Christensen J, Patel JD, Clarke J, Patel SP. Use of Low-Frequency Driver Mutations Detected by Cell-Free Circulating Tumor DNA to Guide Targeted Therapy in Non-Small-Cell Lung Cancer: A

Multicenter Case Series. *JCO Precis Oncol.* 2018;**2**:1-10. doi: 10.1200/PO.17.00318. PubMed PMID: 35135131.

8. Ko TK, Lee E, Ng CC, Yang VS, Farid M, Teh BT, Chan JY, Somasundaram N. Circulating Tumor DNA Mutations in Progressive Gastrointestinal Stromal Tumors Identify Biomarkers of Treatment Resistance and Uncover Potential Therapeutic Strategies. *Front Oncol.* 2022;**12**:840843. Epub 20220222. doi: 10.3389/fonc.2022.840843. PubMed PMID: 35273917; PMCID: PMC8904145.
9. Loeb LA, Kohn BF, Loubet-Seneor KJ, Dunn YJ, Ahn EH, O'Sullivan JN, Salk JJ, Bronner MP, Beckman RA. Extensive subclonal mutational diversity in human colorectal cancer and its significance. *Proc Natl Acad Sci U S A.* 2019;**116**(52):26863-72. Epub 20191205. doi: 10.1073/pnas.1910301116. PubMed PMID: 31806761; PMCID: PMC6936702.
10. Mayo-de-Las-Casas C, Jordana-Ariza N, Garzon-Ibanez M, Balada-Bel A, Bertran-Alamillo J, Viteri-Ramirez S, Reguart N, Munoz-Quintana MA, Lianes-Barragan P, Camps C, Jantus E, Remon-Massip J, Calabuig S, Aguiar D, Gil ML, Vinolas N, Santos-Rodriguez AK, Majem M, Garcia-Pelaez B, Villatoro S, Perez-Rosado A, Monasterio JC, Ovalle E, Catalan MJ, Campos R, Morales-Espinosa D, Martinez-Bueno A, Gonzalez-Cao M, Gonzalez X, Moya-Horno I, Sosa AE, Karachaliou N, Rosell R, Molina-Vila MA. Large scale, prospective screening of EGFR mutations in the blood of advanced NSCLC patients to guide treatment decisions. *Ann Oncol.* 2017;**28**(9):2248-55. doi: 10.1093/annonc/mdx288. PubMed PMID: 28911086.
11. Mizuno K, Sumiyoshi T, Okegawa T, Terada N, Ishitoya S, Miyazaki Y, Kojima T, Katayama H, Fujimoto N, Hatakeyama S, Shiota M, Yoshimura K, Matsui Y, Narita S, Matsumoto H, Kurahashi R, Kanno H, Ito K, Kimura H, Kamiyama Y, Sunada T, Goto T, Kobayashi T, Yamada H, Tsuchiya N, Kamba T, Matsuyama H, Habuchi T, Eto M, Ohyama C, Ito A, Nishiyama H, Okuno H, Kamoto T, Fujimoto A, Ogawa O, Akamatsu S. Clinical Impact of Detecting Low-Frequency Variants in Cell-Free DNA on Treatment of Castration-Resistant Prostate Cancer. *Clin Cancer Res.* 2021;**27**(22):6164-73. Epub 20210915. doi: 10.1158/1078-0432.CCR-21-2328. PubMed PMID: 34526361.
12. Schmitt MW, Loeb LA, Salk JJ. The influence of subclonal resistance mutations on targeted cancer therapy. *Nat Rev Clin Oncol.* 2016;**13**(6):335-47. Epub 20151020. doi: 10.1038/nrclinonc.2015.175. PubMed PMID: 26483300; PMCID: PMC4838548.
13. Shin HT, Choi YL, Yun JW, Kim NKD, Kim SY, Jeon HJ, Nam JY, Lee C, Ryu D, Kim SC, Park K, Lee E, Bae JS, Son DS, Joung JG, Lee J, Kim ST, Ahn MJ, Lee SH, Ahn JS, Lee WY, Oh BY, Park YH, Lee JE, Lee KH, Kim HC, Kim KM, Im YH, Park K, Park PJ, Park WY. Prevalence and detection of low-allele-fraction variants in clinical cancer samples. *Nat Commun.* 2017;**8**(1):1377. Epub 20171109. doi: 10.1038/s41467-017-01470-y. PubMed PMID: 29123093; PMCID: PMC5680209.
14. Thompson JC, Yee SS, Troxel AB, Savitch SL, Fan R, Balli D, Lieberman DB, Morrisette JD, Evans TL, Bauml J, Aggarwal C, Kosteva JA, Alley E, Ciunci C, Cohen RB, Bagley S, Stonehouse-Lee S, Sherry VE, Gilbert E, Langer C, Vachani A, Carpenter EL. Detection of Therapeutically Targetable Driver and Resistance Mutations in Lung Cancer Patients by Next-Generation Sequencing of Cell-Free Circulating Tumor DNA. *Clin Cancer Res.* 2016;**22**(23):5772-82. Epub 20160906. doi: 10.1158/1078-0432.CCR-16-1231. PubMed PMID: 27601595; PMCID: PMC5448134.
15. Topham JT, O'Callaghan CJ, Feilotter H, Kennecke HF, Lee YS, Li W, Banks KC, Quinn K, Renouf DJ, Jonker DJ, Tu D, Chen EX, Loree JM. Circulating Tumor DNA Identifies Diverse Landscape of Acquired Resistance to Anti-Epidermal Growth Factor Receptor Therapy in Metastatic Colorectal Cancer. *J Clin Oncol.* 2023;**41**(3):485-96. Epub 20220825. doi: 10.1200/JCO.22.00364. PubMed PMID: 36007218; PMCID: PMC9870216.

16. Zhang Y, Yao Y, Xu Y, Li L, Gong Y, Zhang K, Zhang M, Guan Y, Chang L, Xia X, Li L, Jia S, Zeng Q. Pan-cancer circulating tumor DNA detection in over 10,000 Chinese patients. *Nat Commun.* 2021;12(1):11. Epub 20210104. doi: 10.1038/s41467-020-20162-8. PubMed PMID: 33397889; PMCID: PMC7782482.
17. Emert BL, Cote CJ, Torre EA, Dardani IP, Jiang CL, Jain N, Shaffer SM, Raj A. Variability within rare cell states enables multiple paths toward drug resistance. *Nat Biotechnol.* 2021;39(7):865-76. Epub 20210222. doi: 10.1038/s41587-021-00837-3. PubMed PMID: 33619394; PMCID: PMC8277666.